# IMITATING GRAPH-BASED PLANNING WITH GOAL-CONDITIONED POLICIES

**Junsu Kim**[1]**, Younggyo Seo**[1]**, Sungsoo Ahn**[2]**, Kyunghwan Son**[1]**, Jinwoo Shin**[1]

[1] Korea Advanced Institute of Science and Technology (KAIST)

[2] Pohang University of Science and Technology (POSTECH)

`{junsu.kim, younggyo.seo, kevinson9473, jinwoos}@kaist.ac.kr`
`sungsoo.ahn@postech.ac.kr`

## ABSTRACT

Recently, graph-based planning algorithms have gained much attention to solve goal-conditioned reinforcement learning (RL) tasks: they provide a sequence of subgoals to reach the target-goal, and the agents learn to execute subgoal-conditioned policies. However, the sample-efficiency of such RL schemes still remains a challenge, particularly for long-horizon tasks. To address this issue, we present a simple yet effective self-imitation scheme which distills a subgoal-conditioned policy into the target-goal-conditioned policy. Our intuition here is that to reach a target-goal, an agent should pass through a subgoal, so target-goal- and subgoal- conditioned policies should be similar to each other. We also propose a novel scheme of stochastically skipping executed subgoals in a planned path, which further improves performance. Unlike prior methods that only utilize graph-based planning in an execution phase, our method transfers knowledge from a planner along with a graph into policy learning. We empirically show that our method can significantly boost the sample-efficiency of the existing goal-conditioned RL methods under various long-horizon control tasks.[1]

## 1 INTRODUCTION

Many sequential decision making problems can be expressed as reaching a given goal, e.g., navigating a walking robot (Schaul et al., 2015; Nachum et al., 2018) and fetching an object using a robot arm (Andrychowicz et al., 2017). Goal-conditioned reinforcement learning (GCRL) aims to solve this problem by training a goal-conditioned policy to guide an agent towards reaching the target-goal. In contrast to many of other reinforcement learning frameworks, GCRL is capable of solving different problems (i.e., different goals) using a single policy.

An intriguing characteristic of GCRL is its *optimal substructure property*; any sub-path of an optimal goal-reaching path is an optimal path for its endpoint (Figure 1a). This implies that a goal-conditioned policy is replaceable by a policy conditioned on a "subgoal" existing between the goal and the agent. Based on this insight, researchers have investigated graph-based planning to construct a goal-reaching path by (a) proposing a series of subgoals and (b) executing policies conditioned on the nearest subgoal (Savinov et al., 2018; Eysenbach et al., 2019; Huang et al., 2019). Since the nearby subgoals are easier to reach than the faraway goal, such planning improves the success ratio of the agent reaching the target-goal during sample collection.

In this paper, we aim to improve the existing GCRL algorithms to be even more faithful to the optimal substructure property. To be specific, we first incorporate the optimal substructure property into the training objective of GCRL to improve the sample collection algorithm. Next, when executing a policy, we consider using all the proposed subgoals as an endpoint of sub-paths instead of using just the subgoal nearest to the agent (Figure 1b).

**Contribution.** We present **P**lanning-guided self-**I**mitation learning for **G**oal-conditioned policies (PIG), a novel and generic framework that builds upon the existing GCRL frameworks that use graph-based planning. PIG consists of the following key ingredients (see Figure 2):

---

[1]Code is available at `https://github.com/junsu-kim97/PIG`

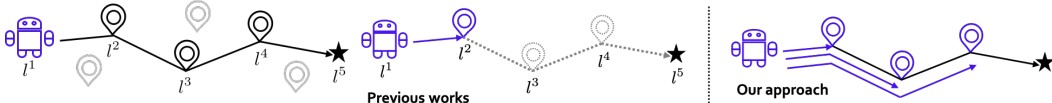

(a) If $(l^1, l^2, l^3, l^4, l^5)$ is an optimal $l^5$-reaching path, all the sub-paths are optimal for reaching $l^5$.

(b) Previous works guide the agent using a $l^2$-reaching sub-path. Our work uses all the possible sub-paths that reach $l^2, l^3, l^4, l^5$.

Figure 1: Illustration of (a) optimal substructure property and (b) sub-paths considered in previous works and our approach for guiding the training of a goal-reaching agent.

- **Training with self-imitation:** we propose a new training objective that encourages a goal-conditioned policy to imitate the subgoal-conditioned policy. Our intuition is that policies conditioned on nearby subgoals are more likely to be accurate than the policies conditioned on a faraway goal. In particular, we consider the imitation of policies conditioned on all the subgoals proposed by the graph-based planning algorithm.

- **Execution[2] with subgoal skipping:** As an additional technique that fits our self-imitation loss, we also propose *subgoal skipping*, which randomizes a subgoal proposed by the graph-based planning to further improve the sample-efficiency. During the sample-collection stage and deployment stage, policies randomly "skip" conditioning on some of the subgoals proposed by the planner when it is likely that the learned policies can reach the proposed subgoals. Such a procedure is based on our intuition that an agent may find a better goal-reaching path by ignoring some subgoals proposed by the planner when the policy is sufficiently trained with our loss.

We demonstrate the effectiveness of PIG on various long-horizon continuous control tasks based on MuJoCo simulator (Todorov et al., 2012). In our experiments, PIG significantly boosts the sample-efficiency of an existing GCRL method, i.e., mapping state space (MSS) (Huang et al., 2019),[3] particularly in long-horizon tasks. For example, MSS + PIG achieves the success rate of 57.41% in Large U-shaped AntMaze environment, while MSS only achieves 19.08%. Intriguingly, we also find that the PIG-trained policy performs competitively even without any planner; this could be useful in some real-world scenarios where planning cost (time or memory) is expensive (Bency et al., 2019; Qureshi et al., 2019).

## 2 RELATED WORK

**Goal-conditioned reinforcement learning (GCRL).** GCRL aims to solve multiple tasks associated with target-goals (Andrychowicz et al., 2017; Kaelbling, 1993; Schaul et al., 2015). Typically, GCRL algorithms rely on the universal value function approximator (UVFA) (Schaul et al., 2015), which is a single neural network that estimates the true value function given not just the states but also the target-goal. Furthermore, researchers have also investigated goal-exploring algorithms (Mendonca et al., 2021; Pong et al., 2020) to avoid any local optima of training the goal-conditioned policy.

**Graph-based planning for GCRL.** To solve long-horizon GCRL problems, graph-based planning can guide the agent to condition its policy on a series of subgoals that are easier to reach than the faraway target goal (Eysenbach et al., 2019; Hoang et al., 2021; Huang et al., 2019; Laskin et al., 2020; Savinov et al., 2018; Zhang et al., 2021). To be specific, the corresponding frameworks build a graph where nodes and edges correspond to states and inter-state distances, respectively. Given a shortest path between two nodes representing the current state and the target-goal, the policy conditions on a subgoal represented by a subsequent node in the path.

For applying graph-based planning to complex environments, recent progress has mainly been made in building a graph that represents visited state space well while being scalable to large environments. For example, Huang et al. (2019) and Hoang et al. (2021) limits the number of nodes in a graph and makes nodes to cover visited state space enough by containing nodes that are far from each other in terms of L2 distance or successor feature similarity, respectively. Moreover, graph sparsification via

---

[2]In this paper, we use the term "execution" to denote both (1) the roll-out in training phase and (2) the deployment in test phase.

[3]We note that PIG is a generic framework that can be also incorporated into any planning-based GCRL methods, other than MSS. Nevertheless, we choose MSS because it is one of the most representative GCRL works as most recent works (Hoang et al., 2021; Zhang et al., 2021) could be considered as variants of MSS.

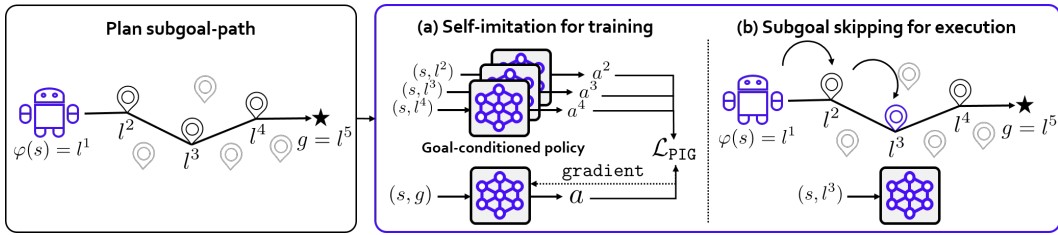

Figure 2: Illustration of **P**lanning-guided self-**I**mitation learning for **G**oal-conditioned policies (PIG). The key ingredient of PIG is twofold: (a) self-imitation for training and (b) subgoal skipping for execution. For (a), we distill a planned-subgoal-conditioned policy into the target-goal-conditioned policy via our self-imitation loss term $\mathcal{L}_{\texttt{PIG}}$. A policy is trained using the auxiliary $\mathcal{L}_{\texttt{PIG}}$ along with off-the-shelf actor loss. For (b), we randomize a subgoal provision from a planner.

two-way consistency (Laskin et al., 2020) or learning latent space with temporal reachability and clustering (Zhang et al., 2021) also have been proposed. They have employed graph-based planning for providing the nearest subgoal to a policy at execution time, which utilizes the optimal substructure property in a limited context. In contrast, PIG aims to faithfully utilize the property both in training and execution via self-imitation and subgoal skipping, respectively.

**Self-imitation learning for goal conditioned policies.** Self-imitation learning strengthens the training signal by imitating trajectories sampled by itself (Oh et al., 2018; Ding et al., 2019; Chane-Sane et al., 2021; Ghosh et al., 2021). GoalGAIL (Ding et al., 2019) imitates goal-conditioned actions from expert demonstrations along with the goal-relabeling strategy (Andrychowicz et al., 2017). Goal-conditioned supervised learning (GCSL) (Ghosh et al., 2021) trains goal-conditioned policies via iterated supervised learning with goal-relabeling. RIS (Chane-Sane et al., 2021) makes target-goal- and subgoal- conditioned policy be similar, where the subgoal is from a high-level policy that is jointly trained with a (low-level) policy. Compared to prior works, PIG faithfully incorporates optimal substructure property with two distinct aspects: (a) graph-based planning and (b) actions from a current policy rather than past actions, where we empirically find that these two differences are important for performance boost (see Section 5.3). Nevertheless, we remark that PIG is an orthogonal framework to them, so applying PIG on top of them (e.g., RIS) would be an interesting future work (e.g., leveraging both planning and high-level policy).

**Distilling planning into a policy.** Our idea of distilling outcomes of planner into the goal-conditioned policy is connected to prior works in the broader planning context. For example, AlphaGo Zero (Silver et al., 2017) distills the outcome of the Monte-Carlo Tree Search (MCTS) planning procedure into a prior policy. Similarly, SAVE (Hamrick et al., 2020) distills the MCTS outcomes into the action-value function. PIG aligns with them in that we distill planned-subgoal-conditioned policy into the target-goal-conditioned policy.

## 3  PRELIMINARY: GOAL-CONDITIONED RL WITH GRAPH-BASED PLANNING

In this section, we describe the existing graph-based planning framework for goal-conditioned reinforcement learning, upon which we build our work. To this end, in Section 3.1, we describe the problem setting of GCRL. Next, in Section 3.2, we explain how to train the goal-conditioned policy using hindsight experience replay (Andrychowicz et al., 2017). Finally, in Section 3.3, we explain how graph-based planning can help the agent to execute better policy. We provide the overall pipeline in Algorithm 1 in Supplemental material A, colored as black.

### 3.1  PROBLEM DESCRIPTION

We formulate our control task as a finite-horizon, goal-conditioned Markov decision process (MDP) (Sutton & Barto, 2018) as a tuple $(\mathcal{S}, \mathcal{G}, \mathcal{A}, p, r, \gamma, H)$ corresponding to state space $\mathcal{S}$, goal space $\mathcal{G}$, action space $\mathcal{A}$, transition dynamics $p(s'|s, a)$ for $s, s' \in \mathcal{S}, a \in \mathcal{A}$, reward function $r(s, a, s', g)$, discount factor $\gamma \in [0, 1)$, and horizon $H$.

Following prior works (Huang et al., 2019; Zhang et al., 2021), we consider a setup where every state can be mapped into the goal space using a goal mapping function $\varphi : \mathcal{S} \to \mathcal{G}$. Then the agent

attempts to reach a certain state $s$ associated with the target-goal $g$, i.e., $\varphi(s) = g$. For example, for a maze-escaping game with continuous locomotion, each state $s$ represents the location and velocity of the agent, while the goal $g$ indicates a certain location desired to be reached by the agent.

Typically, GCRL considers the reward function defined as follows:

$$r(s, a, s', g) = \begin{cases} 0 & \|\varphi(s') - g\|_2 \le \delta \\ -1 & \text{otherwise} \end{cases} \quad (1)$$

where $\delta$ is a pre-set threshold to determine whether the state $s'$ from the transition dynamics $p(s'|s, a)$ is close enough to the goal $g$. To solve GCRL, we optimize a deterministic goal-conditioned policy $\pi :$ $\mathcal{S} \times \mathcal{G} \to \mathcal{A}$ to maximize the expected cumulative future return $V_{g,\pi}(s_0) = \sum_{t=0}^{\infty} \gamma^t r(s_t, a_t, s_{t+1}, g)$ where $t$ denotes timestep and $a_t = \pi(s_t, g)$.

### 3.2 TRAINING WITH HINDSIGHT EXPERIENCE REPLAY

To train goal-conditioned policies, any off-the-shelf RL algorithm can be used. Following prior works (Huang et al., 2019; Zhang et al., 2021), we use deep deterministic policy gradient (DDPG) (Lillicrap et al., 2016) as our base RL algorithm. Specifically, we train an action-value function (critic) $Q$ with parameters $\phi$ and a deterministic policy (actor) $\pi$ with parameters $\theta$ given a replay buffer $\mathcal{B}$, by optimizing the following losses:

$$\mathcal{L}_{\texttt{critic}}(\phi) = \mathbb{E}_{(s_t, a_t, r_t, g) \sim \mathcal{B}}\left[(Q_\phi(s_t, a_t, g) - y_t)^2\right] \quad (2)$$

$$\text{where } y_t = r_t + \gamma Q_\phi(s_{t+1}, \pi_\theta(s_{t+1}, g), g)$$

$$\mathcal{L}_{\texttt{actor}}(\theta) = -\mathbb{E}_{(s_t, a_t, g) \sim \mathcal{B}}[Q_\phi(s_t, \pi_\theta(s_t, g), g)], \quad (3)$$

where the critic $Q_\phi$ is a universal value function approximator (UVFA) (Schaul et al., 2015) trained to estimate the goal-conditioned action-value. However, it is often difficult to train UVFA because the target-goal can be far from the initial position, which makes the agents unable to receive any reward signal. To address the issue, goal-relabeling technique proposed in hindsight experience replay (HER) (Andrychowicz et al., 2017) is widely-used for GCRL methods. The key idea of HER is to reuse any trajectory ending with state $s$ as supervision for reaching the goal $\varphi(s)$. This allows for relabelling any trajectory as success at hindsight even if the agent failed to reach the target-goal during execution.

### 3.3 EXECUTION WITH GRAPH-BASED PLANNING

In prior works (Huang et al., 2019; Zhang et al., 2021), graph-based planning provides a subgoal, which is a waypoint to reach a target goal when executing a policy. A planner runs on a weighted graph that abstracts visited state space.

**Graph construction.** The planning algorithms build a weighted directed graph $\mathcal{H} = (\mathcal{V}, \mathcal{E}, d)$ where each node $l \in \mathcal{V} \subseteq \mathcal{G}$ is specified by a state $s$ visited by the agent, i.e., $l = \varphi(s)$. For populating states, we execute the two-step process following Huang et al. (2019): (a) random sampling of a fixed-sized pool from an experience replay and (b) farthest point sampling (Vassilvitskii & Arthur, 2006) from the pool to build the final collection of landmark states. Then each edge $(l^1, l^2) \in \mathcal{E}$ is assigned for any pair of states that can be visited from one to another by a single transition in the graph. A weight $d(l^1, l^2)$ is an estimated distance between the two nodes, i.e., the minimum number of actions required for the agent to visit node $l^2$ starting from $l^1$. Given $\gamma \approx 1$ and the reward shaped as in Equation 1, one can estimate the distance $d(l^1, l^2)$ as the corresponding value function $-V(s^1, l^2) \approx -Q_\phi(s^1, a^{1,2}, l^2)$ where $l^2 = \varphi(s^2)$ and $a^{1,2} = \pi_\theta(s^1, l^2)$ (Huang et al., 2019). Next, we link all the nodes in the graph and give a weight $d(\cdot, \cdot)$ for each generated edge. Then, if a weight of an edge is greater than (pre-defined) threshold, cut the edge. We provide further details of graph construction in Supplemental material D.1.

**Planning-guided execution.** The graph-based planning provides a policy with an emergent node to visit when executing the policy. To be specific, given a graph $\mathcal{H}$, a state $s$ and, a target goal $g$, we expand the graph by appending $s$ and $g$, and obtain a shortest path $\tau_g = (l^1, \dots, l^N)$ such that $l^1 = \varphi(s)$ and $l^N = g$ using a planning algorithm. Then, a policy is conditioned on a nearby subgoal $l^2$, which is easier to reach than the faraway target-goal $g$. This makes it easy for the agent to collect successful samples reaching the target goals, leading to an overall performance boost. Note that we re-plan for every timestep following prior works (Huang et al., 2019; Zhang et al., 2021).

## 4 PLANNING-GUIDED SELF-IMITATION LEARNING FOR GCRL

In this section, we introduce a new framework, named PIG, for improving the sample-efficiency of GCRL using graph-based planning. Our framework adds two components on top of the existing methods: (a) training with self-imitation and (b) execution with subgoal skipping, which highlights the generality of our concept (colored as purple in Algorithm 1 in Supplementary material A). Our main idea fully leverages the optimal substructure property; any sub-path of an optimal goal-reaching path is an optimal path for its endpoint (Figure 1a). In the following sections, we explain our self-imitation loss as a new training objective in Section 4.1 and subgoal skipping strategy for execution in Section 4.2. We provide an illustration of our framework in Figure 2.

### 4.1 TRAINING WITH SELF-IMITATION

Motivated by the intuition that an agent should pass through a subgoal to reach a target-goal, we encourage actions from target-goal- and subgoal- conditioned policy to stay close, where the subgoals are nodes in a planned subgoal-path. By doing so, we expect that faraway goal-conditioned policy learns plausible actions that are produced by (closer) subgoal-conditioned policy. Specifically, we devise a loss term $\mathcal{L}_{\texttt{PIG}}$ given a stored planned path $\tau_g = (l^1, l^2, \ldots, l^N)$ and a transition $(s, g, \tau_g)$ from a replay buffer $\mathcal{B}$ as follows:

$$\mathcal{L}_{\texttt{PIG}}(\theta) = \mathbb{E}_{(s,\tau_g,g)\sim\mathcal{B}} \left[ \frac{1}{N-1} \sum_{l^k \in \tau_g \setminus \{l^1\}} \| \pi_\theta(s,g) - \texttt{SG}(\pi_\theta(s,l^k)) \|_2^2 \right] \tag{4}$$

where $\texttt{SG}$ refers to a stop-gradient operation. Namely, the goal-conditioned policy imitates behaviors of subgoal-conditioned policy. We incorporate our self-imitation loss term into the existing GCRL frameworks by plugging $\mathcal{L}_{\texttt{PIG}}$ as an extra loss term into the original policy loss term as follows:

$$\mathcal{L}(\theta) = \mathcal{L}_{\texttt{actor}}(\theta) + \lambda \mathcal{L}_{\texttt{PIG}}(\theta) \tag{5}$$

where $\lambda$ is a balancing coefficient, which is a pre-set hyperparameter.

One can also understand that self-imitating loss improves performance by enhancing the correctness of planning. Note that actor is used to estimate distance $d$ between two nodes $l^1, l^2$; $d(l^1, l^2) \approx -Q_\phi(s^1, \pi_\theta(s^1, l^2), l^2)$ as mentioned in Section 3.3. Our self-imitating loss makes $\pi_\theta$ more accurate for even faraway goals, so it leads to the precise construction of a graph. Then, planning gives more suitable subgoals for an actor in execution.

### 4.2 EXECUTION WITH SUBGOAL SKIPPING

As an additional technique that fits our self-imitation loss, we propose *subgoal skipping*, which randomizes a subgoal proposed by the graph-based planning to further improve the sample-efficiency. Note that the existing graph-based planning for GCRL always provides the nearest node $l^2$ in the searched path $\tau_g$ as a desired goal $l^*$ regardless of how a policy is trained. Motivated by our intuition that an agent may find a better goal-reaching path (i.e., short-cuts) by ignoring some of the subgoals, we propose a new subgoal selecting strategy.

Our subgoal skipping is based on the following insight: when a policy for the planned subgoal and the final goal agree (small $\mathcal{L}_{\texttt{PIG}}$), diversifying subgoal suggestions could help find unvisited routes. Namely, the goal-conditioned policy is likely to be trustworthy if final-goal- and planned-subgoal-conditioned policies align because it implies that the goal-conditioned policy have propagated information quite far. Leveraging generalization capability of the trained policy, suggesting the policy with diversified subgoals rather than only the nearest subgoal could help finding better routes.

To be specific, to select the desired goal $l^*$, we start from the nearest node $l^2$ in the planned shortest path $\tau_g$, and stochastically jump to the next node until our condition becomes unsatisfied with the following binomial probability:

$$P[\texttt{jump}] = \min \left( \frac{\alpha}{\mathcal{L}_{\texttt{PIG,latest}}}, 1 \right), \tag{6}$$

where $\alpha$ is pre-set skipping temperature and $\mathcal{L}_{\texttt{PIG,latest}}$ denotes $\mathcal{L}_{\texttt{PIG}}$ calculated at the latest parameter update. We set $l^*$ as the final subgoal after the jumping. Intuitively, the jumping criterion is likely

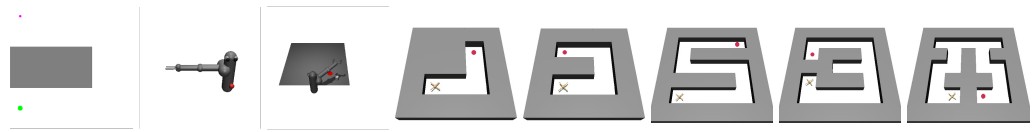

(a) 2DReach (b) Reacher (c) Pusher (d) L-shape (e) U-shape (f) S-shape (g) $\omega$-shape (h) $\Pi$-shape

Figure 3: Environments used in our experiments. In all environments, at training time, an agent starts at a random point, and aims to reach a target goal that is set randomly. At test time for AntMaze tasks, the red point and the position of an ant indicates the target goal, and the initial point, respectively.

to jump more for a smaller $\mathcal{L}_{\texttt{PIG,latest}}$, which is based on the fact that the policy is likely to behave more correctly for a faraway goal. As subgoals are sampled from the searched shortest path and it is not likely to choose a farther subgoal if a policy is not trustworthy for faraway subgoals, our sampled subgoals are likely to be appropriate for a current policy. We describe our subgoal skipping procedure in Algorithm 2 of Supplemental material A.

## 5 EXPERIMENT

In this section, we design our experiments to answer the following questions:

- Can PIG improve the sample-efficiency on long-horizon continuous control tasks over baselines (Figure 4)?
- Can a policy trained by PIG perform well even without a planner at the test time (Figure 5)?
- How does PIG compare to another self-imitation strategy (Figure 6)?
- Is the subgoal skipping effective for sample-efficiency (Figure 7)?
- How does the balancing coefficient $\lambda$ affect performance (Figure 8)?

### 5.1 EXPERIMENTAL SETUP

**Environments.** We conduct our experiments on a set of challenging long-horizon continuous control tasks based on MuJoCo simulator (Todorov et al., 2012). Specifically, we evaluate our framework on 2DReach, Reacher, Pusher, and $\{L, U, S, \omega, \Pi\}$-shaped AntMaze environments (see Figure 3 for the visualization of environments). In 2DReach and AntMaze environments, we use a pre-defined 2-dimensional goal space that represents the $(x, y)$ position of the agent following prior works (Huang et al., 2019; Kim et al., 2021). For Reacher, the goal space is 3-dimension that represents the position of an end-effector. For Pusher, the goal space is 6-dimension that represents the positions of an end-effector and a puck. We provide more details of the environments in Supplemental material B.

**Implementation.** We use DDPG algorithm (Lillicrap et al., 2016) as an underlying RL algorithm following the prior work (Huang et al., 2019). For a graph-based planner and hindsight goal-relabelling strategy, we follow the setup in MSS (Huang et al., 2019). We provide more details of the implementation, including the graph-construcion and hyperparameters in Supplemental material D.

**Evaluation.** We run 10 test episodes without an exploration factor for every 50 training episodes. For the performance metric, we report the success rate defined as the fraction of episodes where the agents succeed in reaching the target-goal within a threshold. We report mean and standard deviation, which are represented as solid lines and shaded regions, respectively, over eight runs for Figure 4 and four runs for the rest of the experiments. For visual clarity, we smooth all the curves equally.

**Baselines and our framework.** We compare our framework with the following baselines on the environments of continuous action spaces:

- HER (Andrychowicz et al., 2017): This method does not use a planner and trains a non-hierarchical policy using a hindsight goal-relabeling strategy.
- MSS (Huang et al., 2019): This method collects samples using a graph-based planner along with a policy and trains the policy using stored transitions with goal-relabeling by HER. A graph is built via farthest point sampling (Vassilvitskii & Arthur, 2006) on states stored in a replay buffer.
- $L^3P$ (Zhang et al., 2021): When building a graph, this method replaces the farthest point sampling of MSS with node-sampling on learned latent space, where nodes are scattered in terms of reachability estimates.

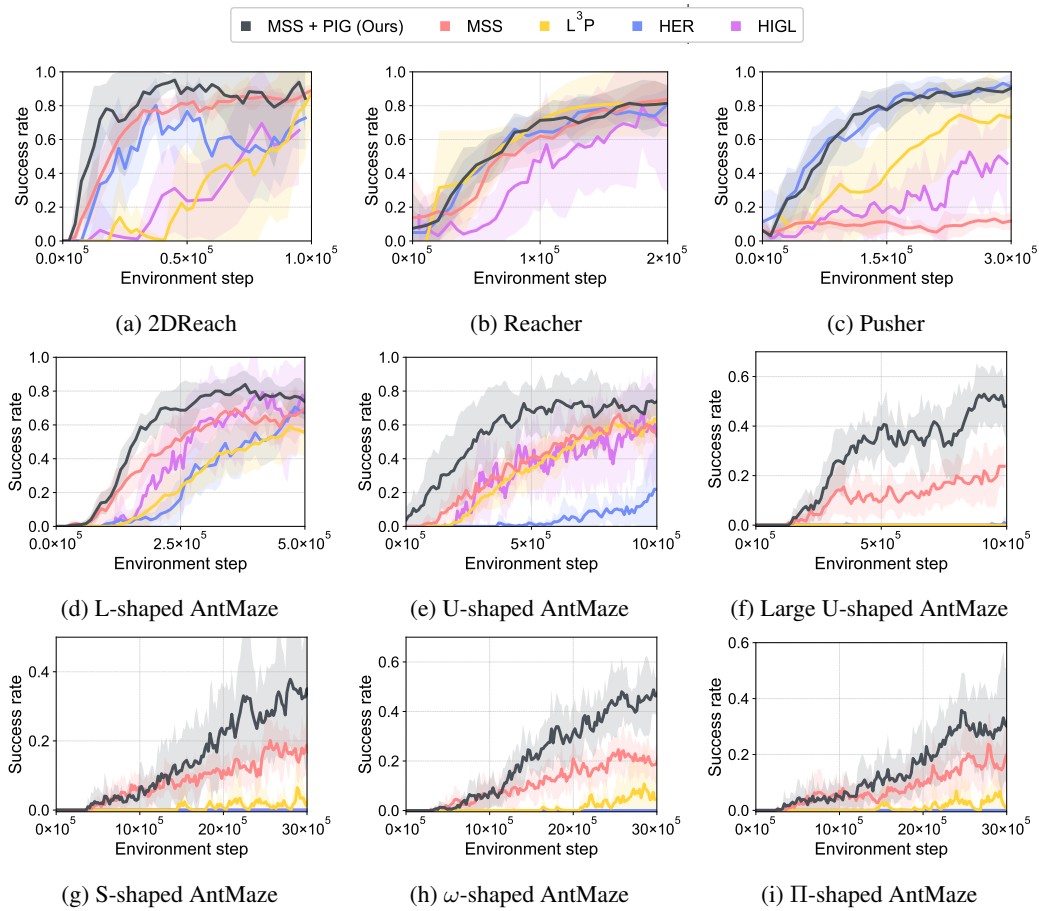

Figure 4: Learning curves on various continuous control tasks as measured on the success rate. We report mean and standard deviation, which are represented as solid lines and shaded regions, respectively, over eight runs. We observe that PIG significantly improves the sample-efficiency of MSS on most tasks. Note that HER and HIGL perform success rate of 0 for Large U-, S-, $\omega$-, and $\Pi$-shaped AntMaze, and $L^3P$ performs success rate of 0 for Large U-shaped AntMaze. GCSL performs success rate of 0 because it does not use planner in execution, which boosts performance in complex long-horizon tasks, so we compare ours with GCSL-variant that uses a planner in Figure 6.

- HIGL (Kim et al., 2021): This method utilizes a graph-based planner to guide training a high-level policy in goal-conditioned hierarchical reinforcement learning. In contrast, PIG uses the planner to guide low-level policy. Comparison with HIGL evaluates the benefits of directly transferring knowledge from the planner to low-level policy without going through high-level policy.

- GCSL (Ghosh et al., 2021): This method learns goal-conditioned policy via iterative supervised learning with goal-relabeling. Originally, GCSL does not utilize a graph-based planner, but we compare ours with GCSL-variant that uses the planner for further investigation in Figure 6.

For all experiments, we report the performance of PIG combined with MSS. Nevertheless, we remark that our work is also compatible with other GCRL approaches because PIG does not depend on specific graph-building or planning algorithms, as can be seen in Algorithm 1 in Supplemental material A. We provide more details about baselines in Supplemental material D.2.

## 5.2 COMPARATIVE EVALUATION

As shown in Figure 4, applying our framework on top of the existing GCRL method, MSS + PIG, improves sample-efficiency with a significant margin across various control tasks. Specifically, MSS + PIG achieves a success rate of 57.41% in large U-shaped AntMaze at environment step $10 \times 10^5$, while MSS performs 19.08%. We emphasize that applying PIG is more effective when the task

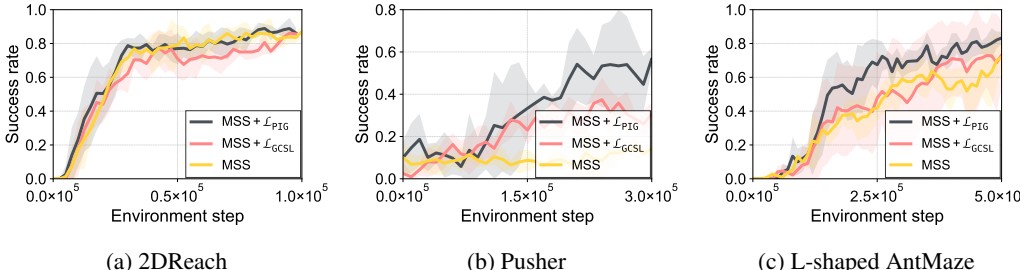

| (a) 2DReach | (b) Pusher | (c) L-shaped AntMaze |

Figure 6: Ablation studies about self-imitation learning for training on (a) 2DReach, (b) Pusher, and (c) L-shaped Ant Maze with four runs. MSS + $\mathcal{L}_{\texttt{PIG}}$ and MSS + $\mathcal{L}_{\texttt{GCSL}}$ refer to an algorithm that applies loss term $\mathcal{L}_{\texttt{PIG}}$ and $\mathcal{L}_{\texttt{GCSL}}$ on top of MSS method, respectively; subgoal skipping is not applied. We find that our loss term $\mathcal{L}_{\texttt{PIG}}$ is more effective than $\mathcal{L}_{\texttt{GCSL}}$ as an auxiliary term.

is more difficult; MSS + PIG shows a larger margin in performance in more difficult tasks (i.e., U-, S-, and $\omega$- shaped mazes rather than L-shaped mazes). Notably, we also observe that MSS + PIG outperforms $L^3P$, which shows that our method can achieve strong performance without the additional complexity of learning latent landmarks. We remark that PIG is also compatible with other GCRL approaches, including $L^3P$, as our framework is agnostic to how graphs are constructed. To further support this, we provide additional experimental results that apply PIG on top of another graph-based GCRL method in Supplemental material C.1.

Also, we find that MSS + PIG outperforms HIGL in Figure 4. These results show that transferring knowledge from a planner to low-level policy is more efficient than passing through a high-level policy. Nevertheless, one can guide both high- and low- level policy via planning, i.e., HIGL + PIG, which would be interesting future work. We also remark that the overhead of applying PIG is negligible in time complexity. Specifically, both the graph-based planning algorithms (MSS+PIG and MSS) spend 1h 30m for 500k steps of the 2DReach, while non-planning baseline (HER) spends 1h.

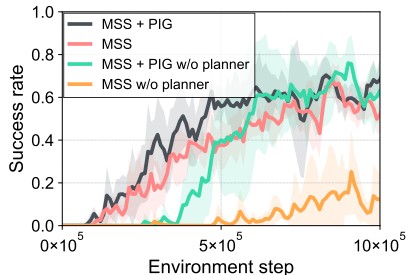

**Reaching a goal without a planner at test time.** To further investigate whether knowledge from graph-based planning is transferred into a policy, we additionally evaluate without the planner at the test time; in other words, the planner is only used at the training time and not anymore at the test time. Intriguingly, we find that training with our PIG enables successfully reaching the target-goal even

Figure 5: Test time success rate of PIG and MSS on U-shaped Ant Maze environment over four runs. The **w/o planner** means that planner is not used at test time, so a goal is directly fed into the policy instead of a subgoal.

without the planner at test time. As shown in Figure 5, this supports our training scheme indeed makes the policy much stronger. Such deployment without planning could be practical in some real-world scenarios where a planning time or memory for storing a graph matter (Bency et al., 2019; Qureshi et al., 2019). We also provide experimental results with a larger maze in Supplemental material C.5.

## 5.3 ABLATION STUDIES

**Effectiveness of our loss design.** In order to empirically demonstrate that utilizing (a) the graph-based planner and (b) actions from a current policy is crucial, we compare PIG (without subgoal skipping) to a GCSL-variant[4] that optimizes the following auxiliary objective in conjunction with the RL objective of MSS framework:

$$\mathcal{L}_{\text{GCSL}} = \mathbb{E}_{(s,a,g)\sim\mathcal{B}}[\|\pi_\theta(s,g) - a\|_2^2], \tag{7}$$

that is, it encourages a goal-conditioned policy to imitate previously successful actions to reach a (relabeled) goal; a goal and a reward is relabeled in hindsight. In execution time, we also apply a graph-based planner to GCSL-variant for a fair comparison. As shown in Figure 6, PIG is more

---

[4]Original GCSL use only $\mathcal{L}_{\text{GCSL}}$, not RL loss term and does not use a planner in execution.

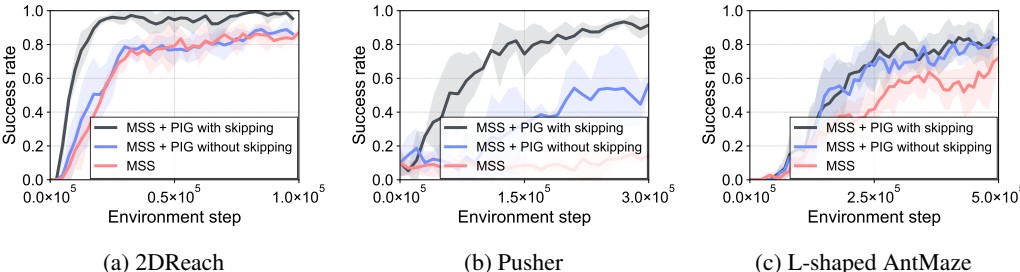

Figure 7: Learning curves of PIG with and without subgoal skipping on (a) 2DReach, (b) Pusher, and (c) L-shaped AntMaze tasks with four runs. PIG with subgoal skipping achieves significantly better performance than without skipping in 2DReach and Pusher.

effective than using the loss $\mathcal{L}_{\text{GCSL}}$ in terms of sample-efficiency due to (a) knowledge transferred by a planner and (b) more plausible actions from a current policy (rather than an old policy).

**Subgoal skipping.** We evaluate whether the proposed subgoal skipping is effective in Figure 7. For 2DReach and Pusher, we observe that PIG with skipping achieves significantly better performance than without skipping. We understand this is because a strong policy may find a better goal-reaching path by ignoring some of the subgoals proposed by the planner. On the other hand, we find that subgoal skipping does not provide a large gain on L-shaped Antmaze, which is a more complex environment. We conjecture that this is because learning a strong policy with high-dimensional state inputs of quadruped ant robots is much more difficult. Nevertheless, we believe this issue can be resolved when the base RL algorithm is more improved. We provide more experiments related to subgoal skipping (i.e., comparison to random skipping) in Supplemental material C.2, C.3, and C.4.

**Balancing coefficient $\lambda$.** We investigate how the balancing coefficient $\lambda$ in Equation 5 that determines the effect of our proposed loss term $\mathcal{L}_{\text{PIG}}$ affect the performance in Figure 8. We find that PIG with $\lambda \in \{1e-3, 1e-4\}$ outperforms PIG with $\lambda = 0$, which shows the importance of the proposed loss. We also observe that too large value of $\lambda$ harms the performance since it incapacitates the training signal of $\mathcal{L}_{\text{actor}}$ excessively. Meanwhile, one can set the balancing coefficient $\lambda$ automatically in a task-agnostic way, which would guide researchers when they extend our work into new environments in the future. We provide experimental results with automatic setting of $\lambda$ in Supplemental material C.3.

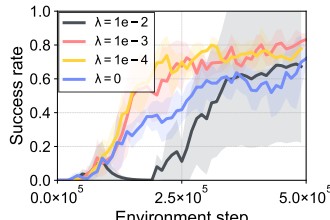

Figure 8: Effectiveness of varying balancing coefficient $\lambda$ on L-shaped AntMaze.

## 6 CONCLUSION

We present PIG, a new self-improving framework that boosts the sample-efficiency in goal-conditioned RL. We remark that PIG is the first work that proposes to guide training and execute with faithfully leveraging the optimal substructure property. Our main idea is (a) distilling planned-subgoal-conditioned policies into the target-goal-conditioned policy and (b) skipping subgoals stochastically in execution based on our loss term. We show that PIG on top of the existing GCRL frameworks enhances sample-efficiency with a significant margin across various control tasks. Moreover, based on our findings that a policy could internalize the knowledge of a planner (e.g., reaching a target-goal without a planner), we expect that such a strong policy would enjoy better usage for the scenarios of transfer learning and domain generalization, which we think an interesting future direction.

**Limitation.** While our experiments demonstrate the PIG on top of graph-based goal-conditioned RL method is effective for solving complex control tasks, we only consider the setup where the state space of an agent is a (proprioceptive) compact vector (i.e., state-based RL) following prior works (Andrychowicz et al., 2017; Huang et al., 2019; Zhang et al., 2021). In principle, PIG is applicable to environments with high-dimensional state spaces because our algorithmic components (self-imitation loss and subgoal skipping) do not depend on the dimensions of state spaces. It would be interesting future work to extend our work into more high-dimensional observation space such as visual inputs. We expect that combining subgoal representation learning (Nachum et al., 2019; Li et al., 2021) (orthogonal methodology to PIG) would be promising.

## REPRODUCIBILITY STATEMENT

We provide the implementation details of our method in Section 5 and Supplemental material D. We also open-source our codebase.

## ETHICS STATEMENT

This work would promote the research in the field of goal-conditioned RL. However, there goal-conditioned RL algorithms could be misused; for example, malicious users could develop autonomous agents that harm society by setting a dangerous goal. Therefore, it is important to devise an method that can take consideration of the consequence of its behaviors to a society.

## ACKNOWLEDGMENTS AND DISCLOSURE OF FUNDING

We thank Sihyun Yu, Jaeho Lee, Jongjin Park, Jihoon Tack, Jaeyeon Won, Woomin Song, Subin Kim, and anonymous reviewers for providing helpful feedbacks and suggestions in improving our paper. This work was supported by Institute of Information & communications Technology Planning & Evaluation (IITP) grant funded by the Korea government(MSIT) (No.2019-0-00075, Artificial Intelligence Graduate School Program(KAIST)). This work was partly supported by Institute of Information & communications Technology Planning & Evaluation (IITP) grant funded by the Korea government(MSIT) (No.2022-0-00953,Self-directed AI Agents with Problem-solving Capability). This work was supported by the National Research Foundation of Korea(NRF) grant funded by the Korea government. (MSIT) (2022R1C1C1013366)

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

## A  ALGORITHM TABLE

We provide algorithm tables that represent PIG in Algorithm 1 and 2.

---

**Algorithm 1** GCRL with planning + PIG

---

**Input**: Number of training episodes $M$, horizon $H$
Initialize replay buffer $\mathcal{B} \leftarrow \varnothing$.
Initialize the parameters of goal-conditioned policy $\pi_\theta$.
Initialize the parameters of action-value function $Q_\phi$.
**for** $m = 1, 2, 3, \ldots M$ **do**
    Reset the environment.
    Sample a target goal $g$ and an initial state $s_0$.
    **for** $t = 1, 2, 3, \ldots H$ **do**
        Build a graph $\mathcal{H} = (\mathcal{V}, \mathcal{E}, d)$ using $\mathcal{B}$.
        Find the shortest subgoal-path $\tau_g$ from $s_t$ to $g$.
        Find a desired subgoal $l^*$ via Algorithm 2.
        Collect a transition $(s_t, a_t, r_t)$ using $\pi_\theta(s_t, l^*)$.
        Store the transition and the planned path $\tau_g$ in $\mathcal{B}$.
    **end for**
    Update $Q_\phi$ using $\mathcal{L}_{\texttt{critic}}(\phi)$ of Equation 2
    Update $\pi_\theta$ using $\mathcal{L}_{\texttt{actor}}(\theta) + \lambda\mathcal{L}_{\texttt{PIG}}(\theta)$ of Equation 5
**end for**

---

**Algorithm 2** Subgoal skipping for execution

---

**Input**: Subgoal-path $\tau_g = (l^1, l^2, \ldots, l^N)$,
        the latest $\mathcal{L}_{\texttt{PIG,latest}}$, normalizing constant $C$.
Initialize desired subgoal $l^* \leftarrow l^2$, current index $i \leftarrow 2$
**while** $i < N$ **do**
    Sample jump according to Equation 6.
    **if** jump **then**
        Update current index $i \leftarrow i + 1$.
        Update desired subgoal $l^* \leftarrow l^i$.
    **else**
        Break while loop.
    **end if**
**end while**
**Output**: desired subgoal $l^*$

---

# B  ENVIRONMENT DETAILS

## B.1  2DREACH

A green point in a 2D U-shaped Maze of size $15 \times 15$ aims to reach a target goal represented by a red point. At each step, the agent can move within $[-1, 1] \times [-1, 1]$ in $x$ and $y$ directions.

## B.2  REACHER

A robotic arm aims to make its end-effector reach the target position on 3D space. The state of the arm is 17-dimension, including the positions, angles, and velocities of itself, and the action-space is 7-dimension. Initial point and target goal are set randomly at the start of episode both at training and test time.

## B.3  PUSHER

A robotic arm aims to make a puck in a plane reach a goal position by pushing the object. The state of the arm is 20-dimension, which is same to Reacher but additionally include position of a puck, and the action-space is 7-dimension. Initial point and target goal are set randomly at the start of episode both at training and test time.

## B.4  ANTMAZE

A quadruped ant robot is trained to reach a random goal from a random location and tested under the most difficult setting for each maze. The states of ant is 30-dimension, including positions and velocities. An ant should reach the target point within 500 steps for U-shaped mazes, and 1000 steps for S-, $\omega$-, and $\Pi$-shaped mazes.

## C ADDITIONAL EXPERIMENTS

### C.1 APPLYING PIG ON TOP OF ANOTHER GRAPH-BASED GCRL METHOD.

Additionally, we also observe that applying PIG on top of another planning-based GCRL method (i.e., $L^3P$ rather than MSS) also demonstrates significant gains. As shown in Figure 9, PIG boost sample-efficiency for $L^3P$ in U-shaped AntMaze and FetchPickAndPlace-v1 (Plappert et al., 2018). These experiments further highlight that PIG is generic technique to improve performance of all the graph-based planning algorithms.

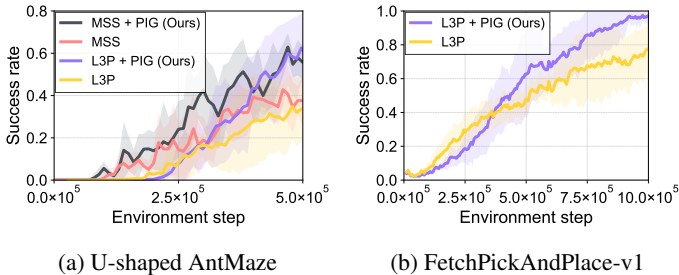

(a) U-shaped AntMaze       (b) FetchPickAndPlace-v1

Figure 9: Test time success rate of PIG on top of another planning-based GCRL method (i.e., $L^3P$) in (a) U-shaped AntMaze and (b) FetchPickAndPlace-v1.

### C.2 COMPARISON TO ALTERNATIVES FOR SUBGOAL SKIPPING.

We compare our subgoal skipping strategy to a simple baseline: random sampling of subgoals from the planned path. As shown in the Figure 10a and 10b, we find that the alternative performs close to ours in 2DReach, but ours outperforms in Pusher. Developing better skipping strategy is an interesting direction to explore.

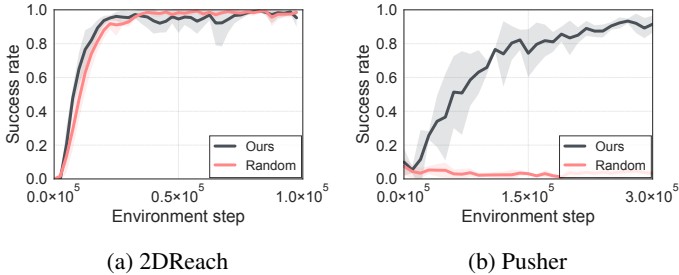

(a) 2DReach       (b) Pusher

Figure 10: Ablation studies about skipping strategy. We compare our skipping strategy to an alternative one: random skipping on (a) 2DReach and (b) Pusher.

### C.3 HYPERPARAMETER TUNING COST.

Our PIG inevitably introduces new hyperparameters ($\lambda$ and $\alpha$) in addition to existing algorithms, but we can use a task-agnostic strategy to choose them without any computational overhead. To be specific, one can set the balancing coefficient $\lambda$ adaptively to satisfy $\lambda \times \mathcal{L}_{\texttt{PIG}} = 0.01 \times \mathcal{L}_{\texttt{actor}}$; see Figure 11a, 11b. Next, we found that the performance of our algorithms is robust to the choice of skipping temperature $\alpha$; see Figure 11c.

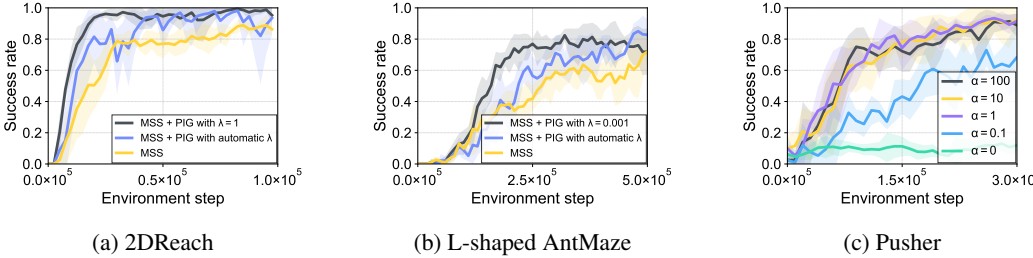

(a) 2DReach      (b) L-shaped AntMaze      (c) Pusher

Figure 11: Experiments with (a, b) automatic hyperparameter setting of $\lambda$ and (c) varying $\alpha$.

## C.4 EFFECT OF SUBGOAL SKIPPING IN EXPLORATION.

To further support our statement - subgoal skipping makes an agent could collect better trajectories via promoting exploration, we quantitatively measure how diverse an agent discovers states during training depending on subgoal skipping. Specifically, we employ particle-based k-nearest neighbors ($k$-NN) entropy estimator (Singh et al., 2003) to measure how diverse collected samples are. Formally, let $X$ be a random variable whose probability density function is $p$, and $\{x_i\}_{i=1}^N$ be its $N$ i.i.d realization. State entropy is defined as $\mathcal{H}(X) = -\mathbb{E}_{x \sim p(x)}[\log p(x)]$ and we can estimate $\mathcal{H}(X)$ as follows:

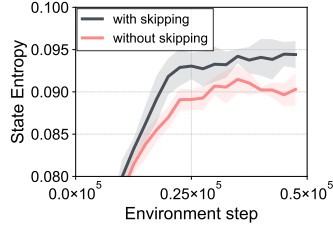

Figure 12: State entropy

$$\hat{\mathcal{H}}_N^K(X) \propto \frac{1}{N} \sum_{i=1}^N \log \frac{1}{K} \sum_{k=1}^K \|x_i - x_i^{k-\text{NN}}\|_2, \tag{8}$$

where $x_i^{k-\text{NN}}$ is the $k$-NN of $x_i$ within a set $\{x_i\}_{i=1}^N$. We use $N = 128$ and $K = 10$ for an experiment using 2DReach environment. As shown in Figure 12, we observe that using subgoal skipping makes high state entropy; that is, subgoal skipping makes an agent collect more diverse samples, which is likely to have more chance to include better samples.

## C.5 REACHING A GOAL WITHOUT A PLANNER AT TEST TIME WITH A LARGER MAZE.

We also evaluate without a planner at test time with a large U-shaped AntMaze. As shown in Figure 13, training with PIG enables successfully reaching the target-goal even without the planner at test time even in larger environment. Intriguingly, after $5 \times 10^5$ environment timesteps, a policy trained by our approach performs better even without access to a planner at test time compared to MSS, which uses a planner at test time.

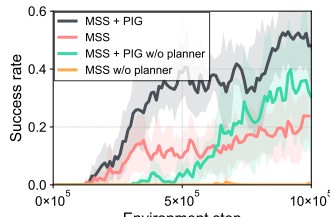

Figure 13: Test time success rate of PIG and MSS on large U-shaped Ant Maze over four runs.

## C.6 Experiments with stochastic transition model.

PIG, along with our graph construction technique, is applicable to stochastic environments since our algorithmic component (self-imitation loss and subgoal skipping) and graph construction mechanism (farthest point sampling and assigning edge weights) are built on visited state spaces, regardless of transition dynamics.

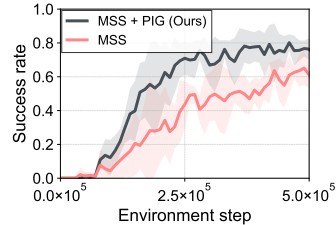

To empirically show that PIG is effective in stochastic environments, we additionally provide experimental results on stochastic L-shaped AntMaze, where gaussian noise $\mathcal{N}(0, 0.05)$ is added to the $(x, y)$ position of an agent at every step following setups from Zhang et al. (2020); Kim et al. (2021). As shown in the Figure 14, we observe that PIG successfully solves tasks in the stochastic environment. Moreover, not only in (deterministic)

Figure 14: Learning curves on stochastic L-shaped AntMaze as measured on the success rate.

L-shaped AntMaze, but also in stochastic L-shaped AntMaze, PIG shows significant gain compared to the baseline (MSS). This result supports that PIG trains a strong policy that is able to reach faraway goals more sample-efficiently than the baseline thanks to our self-imitation loss and subgoal skipping.

## C.7 Ablation studies with more environments.

We provide ablation studies about self-imitation loss and subgoal skipping with more environments: Reacher and Large U-shaped AntMaze. As showin in Figure 15 and 16, including our self-imitation loss or subgoal skipping makes significant gains or performs on par.

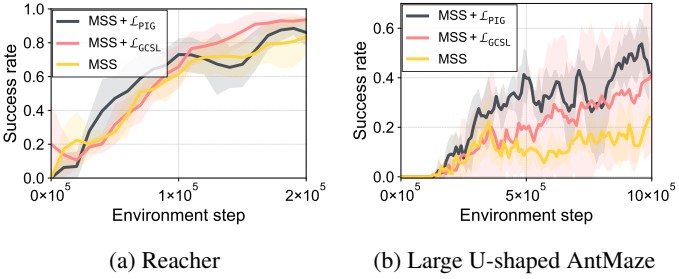

(a) Reacher  (b) Large U-shaped AntMaze

Figure 15: Ablation sutides about self-imitation learning for training on (a) Reacher and (b) Large U-shaped AntMaze with four runs. MSS + $\mathcal{L}_{\texttt{PIG}}$ and MSS + $\mathcal{L}_{\texttt{GCSL}}$ refer to an algorithm that applies loss term $\mathcal{L}_{\texttt{PIG}}$ and $\mathcal{L}_{\texttt{GCSL}}$ on top of MSS method, respectively; subgoal skipping is not applied. We find that our loss term $\mathcal{L}_{\texttt{PIG}}$ is more effective than $\mathcal{L}_{\texttt{GCSL}}$ as an auxiliary term.

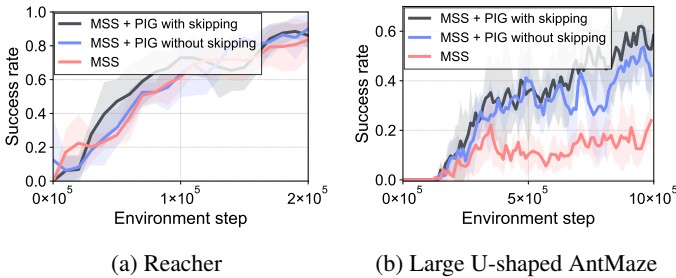

(a) Reacher  (b) Large U-shaped AntMaze

Figure 16: Learning curves of PIG with and without subgoal skipping on (a) Reacher and (b) Large U-shaped AntMaze tasks with four runs.

## C.8 EXPERIMENTS WITH EXTENDED TIMESTEPS.

To assess whether the empirical improvements are in learning speed or also in asymptotic performance, we evaluate PIG and MSS with extended timesteps (i.e., from $10 \times 10^5$ to $30 \times 10^5$ on Large U-shaped AntMaze. As shown in the Table below, we find that PIG can improve both sample-efficiency and asymptotic performances of MSS. This shows that enhanced policy learning via information distillation from the planner can also improve the asymptotic performance.

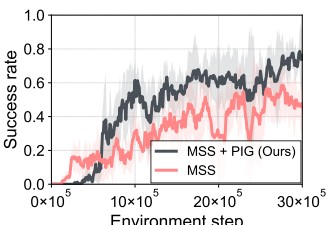

Figure 17: Learning curves on U-shaped AntMaze as measured on the success rate.

## D  IMPLEMENTATION DETAILS

All of the experiments were processed using a single GPU (NVIDIA TITAN Xp) and 8 CPU cores (Intel Xeon E5-2630 v4). For baselines, we employ open-source codes of MSS[5], $L^3P$[6], and HIGL[7].

### D.1  GRAPH CONSTRUCTION

**Collection of graph-constructing states.** We follow collecting scheme of graph-construction states from Huang et al. (2019). The collection is proceeded in two steps: (a) random sampling of a fixed-sized pool $\mathcal{D}$ from an experience replay and (b) farthest point sampling (FPS) (Vassilvitskii & Arthur, 2006; Huang et al., 2019) from the pool $\mathcal{D}$ to build the final collection $\mathcal{V}$ of graph-constructing states.

Specifically, any given time, let $D(s)$ denote the shortest distance from a state $s$ to the closest element in current $\mathcal{V}$. The set $\mathcal{V}$ is initialized with an empty set. Then, FPS runs as follows:

- Step A: Choose a state $s^1$ uniformly at random from the pool $\mathcal{D}$ and add $s^1$ into $\mathcal{V}$.
- Step B: Choose the next state $s^i$, whose $D(s^i)$ is the largest among elements in $\mathcal{D}$. Add $s^i$ into $\mathcal{V}$.
- Step C: Repeat Step B until we have chosen a budget for the number of nodes in a graph.

The diversity of the collection is ensured by farthest point sampling. Random sampling to build a fixed-sized pool makes the computational complexity of planning irrelevant to the size of experience replay, of which size is 1M in our experiments.

**Edge connection.** After collecting graph-constructing states, we complete a graph by adding directed edges (Huang et al., 2019). In detail, given two nodes $l^1$ and $l^2$, we connect them by adding two directed edges $(l^1, l^2) \in \mathcal{E}$ (from $l^1$ to $l^2$) and $(l^2, l^1) \in \mathcal{E}$ (from $l^2$ to $l^1$). Then we assign weights as an estimated distance $d(l^1, l^2)$ and $d(l^2, l^1)$, respectively.

### D.2  HYPERPARAMETERS

We list hyperparameters used for PIG across all environments in Table 1 and 2.

For the baselines, we used the best hyperparameters reported in their source codes for shared environments: 2DReach of MSS and HER, Reacher and Pusher for HIGL, and AntMazes for MSS, L3P, HER, and HIGL (all). For unstudied environments in the baseline papers, we have searched hyperparameters for each baseline. For example, we search shift magnitude and adjacency degree for HIGL, clipping threshold and final goal adjacency threshold for L3P and MSS, and relabeling ratio for HER. We note that for PIG, two newly introduced hyperparameters (balancing coefficient $\lambda$ and skipping temperature $\alpha$) have been searched. We would like to remark that performance

---

[5]https://github.com/FangchenLiu/map_planner
[6]https://github.com/LunjunZhang/world-model-as-a-graph
[7]https://github.com/junsu-kim97/HIGL

gain by PIG have been achieved without exhaustive efforts in hyperparameter search compared to baselines. For example, the baseline MSS conducted grid search on 30 (number of landmarks) $\times$ 30 (clipping threshold) values in their paper, but we searched among $5 \times 4$ values for PIG: $\{1.0, 0.1, 0.01, 0.001, 0.0001\}$ for $\lambda$ and $\{20, 10, 5, 1\}$ for $\alpha$.

Table 1: Hyperparameters across all environments.

| Hyperparameter | Value |
|---|:---:|
| *DDPG* | |
| Optimizer | Adam (Kingma & Ba, 2014) |
| Actor learning rate | 0.0002 |
| Critic learning rate | 0.0002 |
| Replay buffer size | 1M |
| Number of hidden layers for actors | 4 |
| Number of hidden layers for critics | 5 |
| Number of hidden units per layer | 400 |
| Batch size | 200 |
| Nonlinearity | ReLU |
| Polyak for target network | 0.99 |
| Target update frequency per episode | 3 |
| Ratio between env vs optimization steps | 1 |
| Gamma | 0.99 |
| Hindsight relabelling ratio | 0.8 |
| *Graph* | |
| Number of soft value iteration | 20 |
| Temperature | 0.9 |

Table 2: Hyperparameters that differ across the environments.

| Hyperparameter | 2DReach | Reacher | Pusher | AntMaze |
|---|:---:|:---:|:---:|:---:|
| *Ours-specific* | | | | |
| Balancing coefficient $\lambda$ | 1.0 | 0.0001 | 0.1 | 0.001 |
| Skipping temperature $\alpha$ | 1.0 | 10.0 | 1.0 | 10.0 |
| *DDPG* | | | | |
| Initial random trajectories | 2.5k | 20k | 20k | 100k (for L-, U- shaped Maze) 400k (for Large U-shaped Maze) 800k (for S-, $\omega$-, $\Pi$ -shaped Maze) |
| Hindsight relabelling range | 50 | 50 | 50 | 200 |
| Action L2 | 0.5 | 0.01 | 0.01 | 0.5 |
| Action noise | 0.2 | 0.1 | 0.1 | 0.2 |
| *Graph* | | | | |
| Number of nodes in a graph | 100 | 80 | 80 | 400 |
| clipping threshold for distances | 4.0 | 4.0 | 4.0 | 38.0 |

