# OpenReview forum: "Imitating Graph-Based Planning with Goal-Conditioned Policies"
_ICLR.cc/2023/Conference — ICLR 2023 poster_

### Official Review · Reviewer_syAR · 2022-10-21

**Confidence:** 4
**Correctness:** 3
**Technical Novelty And Significance:** 2
**Empirical Novelty And Significance:** 2
**Recommendation:** 6

**Clarity, Quality, Novelty And Reproducibility:**

Clarity: Good
Quality: Decent
Novelty: Moderate
Reproducibility: Should be reproducible.

**Strength And Weaknesses:**

Strength:
* The paper is clearly written.
* Related work is well covered.
* Section 3 is a very clear introduction to the overall setting and previous approaches.
* Good ablation experiments.

Weaknesses:
* My biggest issue is with the evaluation of the baselines. You indicate that for the baselines you “used the best hyperparameters reported in their source codes”. However, the baseline papers do not study all the environments you use, and are often optimized for other environments. This can make your comparison (very) unfair. If you optimized you own hyperparameters, but left their hyperparameters untouched (which were optimized for other tasks), then this will strongly influence performance. I now miss details about this procedure, and in principle I think you should put as much hyperparameter optimization effort in the baselines as you put into your own method.
* Subgoal skipping gets a lot of attention in your introduction and methodology, but then in the experiments it only features in Fig 7 and a single paragraph of results.
* In the second part of your results (Fig 6 and 7) you suddenly reduce the set of environments to three. How did you choose these three (why do you show these ones, and not the other ones?). You need to tell us why these are not cherry-picked.
* The contribution is slightly incremental. Most focus lies on the self-imitation loss. I do think this is a nice insight, but effectively it adds a single loss term to a known algorithm. It is still an interesting idea, but not totally ground-breaking.

Minor:
* Sec 2: You say that progress lies in “compressing the graph by removing redundant nodes”, but then you give examples like “semi-parametric topological memory”, “farthest point sampling” and “successor features”, which do not say anything about how you reduce the number of nodes in the subgoal graph.
* Sec 3.3: You need to explain “farthest point sampling”.
* Sec 4.2: * I had to read Sec 4.2 multiple times to understand your motivation for Eq 6, I think you could write this down more clearly. I guess it says that, when the policy for the current subgoal and the final goal agree (small L_PIG), the goal-conditioned policy has propagated information quite far. Therefore, we may try to jump to further ahead subgoals, because the goal-conditioned policy seems trustworthy. I think you could phrase this more explicitly.


**Summary Of The Paper:**

This paper studies goal-conditioned RL, in particular the graph-based approach, where we build a graph of subgoals to improve overall performance. Compared to previous methods, this paper introduces two improvements: 1) an additional loss term to make subgoal policies compatible with end goal policies, and 2) at execution time a stochastic subgoal skipping mechanism which makes the agent sometimes aim for further ahead subgoals in the planned path. Experiments in a range of tasks show improvement on some tasks, while being on par with baselines on other environments.

**Summary Of The Review:**

The paper is well written and clear, has extensive experiments, and introduces a method that manages to outperform baselines on most of the tasks. My biggest issue is the experimental set-up, where the authors have optimized the hyperparameters of their own algorithm, but reused hyperparameters of the baselines that were optimized for different tasks. The contribution is slightly incremental (two tweaks to a known algorithm), but still interesting. I'm on a borderline vote for this paper.

---

> ### Author Response · Authors · 2022-11-11
> **Response to reviewer syAR (1/2)**
>
> Dear reviewer syAR,
>
> We express our deep appreciation for your time and insightful comments. We address your comments one by one. The revisions made are marked with “$\text{\color{magenta}magenta}$” in the revised paper.
>
> ---
>
> **Q1: My biggest issue is with the evaluation of the baselines. You indicate that for the baselines you “used the best hyperparameters reported in their source codes”. However, the baseline papers do not study all the environments you use, and are often optimized for other environments. This can make your comparison (very) unfair. If you optimized you own hyperparameters, but left their hyperparameters untouched (which were optimized for other tasks), then this will strongly influence performance. I now miss details about this procedure, and in principle I think you should put as much hyperparameter optimization effort in the baselines as you put into your own method.**
>
> **A1.** We used the best hyperparameters reported in their source codes for shared environments: 2DReach of MSS and HER, Reacher and Pusher for HIGL, and AntMazes for MSS, L3P, HER, and HIGL (all). For unstudied environments in the baseline papers, we have searched hyperparameters for each baseline. For example, we search shift magnitude and adjacency degree for HIGL, clipping threshold and final goal adjacency threshold for L3P and MSS, and relabeling ratio for HER. We note that for PIG, two newly introduced hyperparameters (balancing coefficient $\lambda$ and skipping temperature $\alpha$) have been searched. We would like to remark that performance gain by PIG has been achieved without exhaustive efforts in hyperparameter search compared to baselines. For example, the baseline MSS conducted a grid search on 30 (number of landmarks) $\times$ 30 (clipping threshold) values in their paper, but we searched among 5 ($\lambda$) $\times$ 4 ($\alpha$) values for PIG: {1.0, 0.1, 0.01, 0.001, 0.0001} for $\lambda$ and {20, 10, 5, 1} for $\alpha$. We clarify this in Supplementary D.2 of the revised paper.
>
> ---
>
> **Q2: Subgoal skipping gets a lot of attention in your introduction and methodology, but then in the experiments it only features in Fig 7 and a single paragraph of results.**
>
> **A2.** For subgoal skipping-related experiments, we have conducted not only an ablation study for subgoal skipping (Figure 7) but also a comparison to random subgoal skipping (Figure 10), the effectiveness of skipping temperature (Figure 11-c), and the quantitative effect of subgoal skipping in exploration (Figure 12) in our supplementary material of original draft. In the revised paper, we add a guiding sentence in the main text for readers to easily find subgoal skipping related experiments in the Supplemental material C.2, C.3, and C.4.
>
> ---
>
> **Q3: In the second part of your results (Fig 6 and 7) you suddenly reduce the set of environments to three. How did you choose these three (why do you show these ones, and not the other ones?). You need to tell us why these are not cherry-picked.**
>
> A3: In the original draft, we chose three environments with different characteristics: 2DReach for 2d navigation, Pusher for robot manipulation, and L-shaped AntMaze for navigation with locomotion. To further address your comment, we provide results with more environments (Reacher and large U-shaped AntMaze). As shown in the tables below, including our self-imitation loss or subgoal skipping makes significant gains or performs on par. We include these additional experimental results in Supplemental material C.7 of the revised paper.
>
> ###### $\textbf{Mean (standard deviation) of success rate on Reacher}$
> \begin{array}{lcccc}
> \text{Timesteps}  & 0.5 * 10^5 & 1.0 * 10^5 & 1.5 * 10^5 & 2.0 * 10^5 \newline
> \hline
> \\textbf{MSS +} L_{\\text{\textbf{PIG}}} & \bf{0.47} \\; (0.13) & \bf{0.73} \\; (0.10) & 0.67 \\; (0.14) & 0.86 \\; (0.08) & \newline
> \\text{MSS +} L_{\\text{GCSL}} & 0.31 \\; (0.12) & 0.66 \\; (0.13) & \bf{0.87} \\; (0.09) & \bf{0.94} \\; (0.02) & \newline
> \\text{MSS} & 0.27 \\; (0.07) & 0.62 \\; (0.08) & 0.71 \\; (0.12) & 0.83 \\; (0.11) \newline
> \end{array}
>
> ###### $\textbf{Mean (standard deviation) of success rate on U-shaped AntMaze}$
> \begin{array}{lcccc}
> \text{Timesteps}  & 2.5 * 10^5 & 5.0 * 10^5 & 7.5 * 10^5 & 10.0 * 10^5 \newline
> \hline
> \\textbf{MSS +} L_{\\text{\textbf{PIG}}} & 0.08 \\; (0.03) & \bf{0.40} \\; (0.11) & 0.29 \\; (0.14) & \bf{0.42} \\; (0.11) \newline
> \\text{MSS +} L_{\\text{GCSL}} & \bf{0.09} \\; (0.14) & 0.19 \\; (0.22) & \bf{0.30} \\; (0.31) & 0.39 \\; (0.30) \newline
> \\text{MSS} & 0.03 \\; (0.04) & 0.10 \\; (0.07) & 0.17 \\; (0.08) & 0.24 \\; (0.12) & \newline
> \end{array}

---

> > ### Author Response · Authors · 2022-11-11
> > **Response to reviewer syAR (2/2)**
> >
> > ###### $\textbf{Mean (standard deviation) of success rate on Reacher}$
> > \begin{array}{lcccc}
> > \text{Timesteps}  & 0.5 * 10^5 & 1.0 * 10^5 & 1.5 * 10^5 & 2.0 * 10^5 \newline
> > \hline
> > \\textbf{MSS +} L_{\\text{\textbf{PIG}}} & \bf{0.47} \\; (0.13) & \bf{0.73} \\; (0.10) & 0.67 \\; (0.14) & 0.86 \\; (0.08) & \newline
> > \\text{MSS +} L_{\\text{GCSL}} & 0.32 \\; (0.11) & 0.63 \\; (0.14) & \bf{0.85} \\; (0.03) & \bf{0.90} \\; (0.08) \newline
> > \\text{MSS}  & 0.03 \\; (0.04) & 0.10 \\; (0.07) & 0.17 \\; (0.08) & 0.24 \\; (0.12) \newline
> > \end{array}
> >
> > ###### $\textbf{Mean (standard deviation) of success rate on U-shaped AntMaze}$
> > \begin{array}{lcccc}
> > \text{Timesteps}  & 2.5 * 10^5 & 5.0 * 10^5 & 7.5 * 10^5 & 10.0 * 10^5 \newline
> > \hline
> > \\textbf{MSS +} L_{\\text{\textbf{PIG}}} & \bf{0.15} \\; (0.14) & 0.36 \\; (0.21) & \bf{0.51} \\; (0.15) & \bf{0.59} \\; (0.15) \newline
> > \\text{MSS +} L_{\\text{GCSL}} & 0.08 \\; (0.03) & \bf{0.40} \\; (0.11) & 0.29 \\; (0.14) & 0.42 \\; (0.11) & \newline
> > \\text{MSS} & 0.03 \\; (0.04) & 0.10 \\; (0.07) & 0.17 \\; (0.08) & 0.24 \\; (0.12)\\newline\end{array}
> >
> > ---
> >
> > **Q4: The contribution is slightly incremental. Most focus lies on the self-imitation loss. I do think this is a nice insight, but effectively it adds a single loss term to a known algorithm. It is still an interesting idea, but not totally ground-breaking.**
> >
> > **A4.** We would like to emphasize that our main contribution is being the first to consider optimal substructure property for GCRL. Our self-imitation loss and subgoal skipping are our technical contributions to enforce optimal substructure property for GCRL. We also believe the simplicity of our algorithm implies that future researchers can easily build up on our idea and propose more sophisticated algorithms.
> >
> > ---
> >
> > **Q5: Minor comments: Sec 2: You say that progress lies in “compressing the graph by removing redundant nodes”, but then you give examples like “semi-parametric topological memory”, “farthest point sampling” and “successor features”, which do not say anything about how you reduce the number of nodes in the subgoal graph.**
> >
> > **A5.** Thank you for the comment. Semi-parametric topological memory [1] does not compress a graph, so this may become problematic in very large environments. To mitigate this problem, prior works used farthest point sampling [2] and successor feature [3] to limit the number of nodes in a graph and aimed at containing nodes that are far from each other in terms of L2 distance or successor feature similarity, respectively. In Section 2 of the revised paper, we re-phrase the sentences and explain with more details to help readers catch up with the prior works.
> >
> > **References**
> >
> > [1] Nikolay Savinov, Alexey Dosovitskiy, and Vladlen Koltun. “Semi-parametric topological memory for navigation”, ICLR 2018.
> >
> > [2] Zhiao Huang, Fangchen Liu, and Hao Su. “Mapping state space using landmarks for universal goal reaching”, NeurIPS 2019.
> >
> > [3] Christopher Hoang, Sungryull Sohn, Jongwook Choi, Wilka Carvalho, and Honglak Lee. Successor feature landmarks for long-horizon goal-conditioned reinforcement learning. NeurIPS 2021.
> >
> > ---
> >
> > **Q6: Minor comments:  Minor: Sec 3.3: You need to explain “farthest point sampling”.**
> >
> > **A6.** To address your comment, we explain farthest point sampling in detail in Supplemental material D of the revised paper. Specifically, given a fixed size of pool $\mathcal{D}$, we build the final collection $\mathcal{V}$ of graph-constructing states via farthest point sampling (FPS). Let $D(s)$ denote the shortest distance from a state $s$ to the closest element in current $\mathcal{V}$. The set $\mathcal{V}$ is initialized with an empty set. Then, FPS runs as follows:
> > - Step A: Choose a state $s^{1}$ uniformly at random from the pool $\mathcal{D}$ and add $s^{1}$ into $\mathcal{V}$.
> > - Step B: Choose the next state $s^{i}$, whose $D(s^{i})$ is the largest among elements in $\mathcal{D}$. Add $s^{i}$ into $\mathcal{V}$.
> > - Step C: Repeat Step B until we have chosen a budget for the number of nodes in a graph.
> >
> > ---
> >
> > **Q7: Minor comments:  Minor: Sec 4.2: * I had to read Sec 4.2 multiple times to understand your motivation for Eq 6, I think you could write this down more clearly. I guess it says that when the policy for the current subgoal and the final goal agree (small L_PIG), the goal-conditioned policy has propagated information quite far. Therefore, we may try to jump to further ahead subgoals, because the goal-conditioned policy seems trustworthy. I think you could phrase this more explicitly.**
> >
> > **A7.** Following your comment, we revise the corresponding paragraph in Section 4.2 to help a clearer understanding. Thank you!

---

### Official Review · Reviewer_X3ba · 2022-10-23

**Confidence:** 4
**Clarity, Quality, Novelty And Reproducibility:** The paper is clear, but the novelty i…
**Correctness:** 4
**Technical Novelty And Significance:** 2
**Empirical Novelty And Significance:** 2
**Recommendation:** 6

**Strength And Weaknesses:**

The paper is well-written and organized, and the proposed method is simple but seems to be effective. However, the paper may not have a high impact on reinforcement learning due to some limitations of PIG.

PIG seems to be effective only for goal-reaching tasks, and all the experiments are conducted in goal-reaching tasks. Whether is PIG applied to tasks with complex reward functions?

Whether is the proposed graph construction technique applied to complex environments with stochastic transition models, directed-graph-based, or high-dimensional state spaces?

Subgoal skipping may induce a farther subgoal that is difficult to reach by the agent. How to guarantee that the technique can provide appropriate subgoals for the agent?


**Summary Of The Paper:**

The paper presents a novel and generic framework Planning-guided self-Imitation learning for Goal-conditioned policies (PIG) to improve the sample efficiency of goal-conditioned RL (GCRL). Empirical results show that PIG significantly boosts the performance of the existing GCRL methods under various goal-reaching tasks.

**Summary Of The Review:**

The current version does not convince me to recommend acceptance


######After Rebuttal######

The additional explanation and experiments address most of my concerns

---

> ### Author Response · Authors · 2022-11-11
> **Response to reviewer X3ba (1/2)**
>
> Dear reviewer X3ba,
>
> We express our deep appreciation for your time and insightful comments. We address your comments one by one. The revisions made are marked with “$\text{\color{magenta}magenta}$” in the revised paper.
>
> ---
>
> **Q1: PIG seems to be effective only for goal-reaching tasks, and all the experiments are conducted in goal-reaching tasks. Whether is PIG applied to tasks with complex reward functions?**
>
> **A1.** PIG is a goal-conditioned reinforcement learning (GCRL) algorithm, and every GCRL algorithm focus on solving goal-reaching tasks; if a task is specified with only a (complex) reward function and the final goal is not provided, GCRL approaches, as well as PIG, cannot be applied to the task. However, we remark that the field of GCRL is significant since (1) providing a final goal is often cheaper than designing a complex reward function, which requires lots of domain knowledge, and (2) many important real-world problems (i.e., navigation and manipulation) can be formulated as a goal-reaching problem. Hence, we believe that PIG, as well as GCRL, would be favorable for many practitioners.
>
> ---
>
> **Q2: whether is the proposed graph construction technique applied to complex environments with stochastic transition models, directed-graph-based, or high-dimensional state spaces?**
>
> **A2.** PIG, along with our graph construction technique, can be applied to (1) stochastic transition models, (2) a directed graph, and (3) high-dimensional state space. In the following paragraphs, we elaborate on how our approach can be applied to them.
> - $\textbf{Stochastic transition model.}$ PIG, along with our graph construction technique, is applicable to stochastic environments since our algorithmic component (self-imitation loss and subgoal skipping) and graph construction mechanism (farthest point sampling and assigning edge weights) are built on visited state spaces, regardless of transition dynamics. To empirically show that PIG is effective in stochastic environments, we additionally provide experimental results on stochastic L-shaped AntMaze, where gaussian noise $\mathcal{N}(0, 0.05)$ is added to the $(x, y)$ position of an agent at every step following setups from [1, 2]. As shown in the Table below, we observe that PIG successfully solves tasks in stochastic environments. Moreover, not only in (deterministic) L-shaped AntMaze but also in stochastic L-shaped AntMaze, PIG shows significant gain compared to the baseline (MSS). This result supports that PIG trains a strong policy that is able to reach faraway goals more sample-efficiently than the baseline, thanks to our self-imitation loss and subgoal skipping. We add the results and related discussion in Supplemental material C.6 in the revised paper.
>
> ###### $\textbf{Mean (standard deviation) of success rate on stochastic L-shaped AntMaze}$
> \begin{array}{lccccc}
> \text{Timesteps}  & 1.0 * 10^5 & 2.0 * 10^5 & 3.0 * 10^5 & 4.0 * 10^5 & 5.0 * 10^5 \newline
> \hline
> \textbf{MSS + PIG (Ours)} & \bf{0.12} \\; (0.11) & \bf{0.56} \\; (0.22) & \bf{0.67} \\; (0.12) & \bf{0.78} \\; (0.10) & \bf{0.76} \\; (0.06) \newline
> \text{MSS} & 0.04 \\; (0.06) & 0.29 \\; (0.20) & 0.48 \\; (0.14) & 0.49 \\; (0.09) & 0.61 \\; (0.07) \newline
> \end{array}
>
> - $\textbf{Directed graph.}$ we clarify that our algorithm already runs on top of a “directed”-graph; each edge $(l^{1}, l^{2}) \in \mathcal{E}$ has a direction from a node $l^{1}$ to another node $l^{2}$ [3]. In detail, when we build an edge given two nodes $l^{1}$ and $l^{2}$, we connect them by adding two directed edges $(l^{1}, l^{2})$ (i.e., from $l^{1}$ to $l^{2}$) and $(l^{2}, l^{1})$ (i.e., from $l^{2}$ to $l^{1}$). Then we assign weights as an estimated distance $d(l^{1}, l^{2})$ and $d(l^{2}, l^{1})$, respectively. We further clarify that PIG runs on top of the “directed”-graph in Section 3.3 and Supplemental material D.1 of the revised paper.

---

> > ### Author Response · Authors · 2022-11-11
> > **Response to reviewer X3ba (2/2)**
> >
> > - $\textbf{High-dimensional state spaces.}$ In principle, PIG is applicable to environments with high-dimensional state spaces because our algorithmic components (self-imitation loss and subgoal skipping) do not depend on the dimensions of state spaces. To further improve performance in high-dimensional state spaces, we expect that combining subgoal representation learning [4, 5] (orthogonal methodology to PIG) would be promising. We add the related discussion in Section 6 of the revised paper.
> >
> > **References**
> >
> > [1] Tianren Zhang, Shangqi Guo, Tian Tan, Xiaolin Hu, Feng Chen, “Generating Adjacency-Constrained Subgoals in Hierarchical Reinforcement Learning”, NeurIPS 2020.
> >
> > [2] Junsu Kim, Younggyo Seo, Jinwoo Shin, “Landmark-Guided Subgoal Generation in Hierarchical Reinforcement Learning”, NeurIPS 2021.
> >
> > [3] Zhiao Huang, Fangchen Liu, Hao Su, “Mapping State Space using Landmarks for Universal Goal Reaching”, NeurIPS 2019.
> >
> > [4] Ofir Nachum, Shixiang Gu, Honglak Lee, Sergey Levine, “Near-optimal representation learning for hierarchical reinforcement learning”, ICLR 2019.
> >
> > [5] Siyuan Li, Lulu Zheng, Jianhao Wang, Chongjie Zhang, “Learning subgoal representations with slow dynamics”, ICLR 2021.
> >
> > ---
> >
> > **Q3: Subgoal skipping may induce a farther subgoal that is difficult to reach by the agent. How to guarantee that the technique can provide appropriate subgoals for the agent?**
> >
> > **A3.** Our subgoal skipping strategy provides appropriate subgoals because (1) subgoals are sampled from the searched shortest path, and (2) it is not likely to choose a farther subgoal if a policy is not trustworthy for faraway subgoals. Specifically, for (1), all the sampled subgoals are on the way to the final goal, so they are proper waypoints to achieve the final goal. For (2), thanks to our skipping design, we rarely skip subgoals if a policy is not trustworthy for faraway subgoals. We enrich Section 4.2 of the revised paper with this discussion.

---

> > > ### Comment · Reviewer_X3ba · 2022-11-14
> > > **Response**
> > >
> > > Thanks for the additional explanation and experiments that address most of my concerns. I would like to raise the score to six.
> > >
> > > I agree that all GCRL algorithms including PIG focus on solving goal-reaching tasks. Nevertheless, I believe applying only to goal-reaching tasks is the biggest issue because many RL tasks do not have an explicit goal. Please add this discussion as well as the above explanation and experiments in the final versoin.

---

> > > > ### Author Response · Authors · 2022-11-15
> > > > **Thank you for the response**
> > > >
> > > > We are happy to hear that our rebuttal addressed your concerns well.
> > > >
> > > > We will add the suggested discussion as well as the above explanation and experiments into the final version, which we believe further strengthen our paper !
> > > >
> > > > Best, Authors.

---

### Official Review · Reviewer_4xSo · 2022-10-24

**Confidence:** 3
**Correctness:** 3
**Technical Novelty And Significance:** 3
**Empirical Novelty And Significance:** 3
**Recommendation:** 8

**Clarity, Quality, Novelty And Reproducibility:**

* Clarity: The paper is well-written and easy to follow. The methods are clearly presented.
* Quality: The provided method is solid. Experiments are well designed and results are thoroughly discussed.
* Novelty: to the best of my knowledge, the proposed two techniques in PIG are novel.
* Reproducibility: Sourcescode is provided. Implementation details are described in the appendix.

**Strength And Weaknesses:**

# Strength
* The paper is well-written and easy to follow.
* The ideas of the two techniques in PIG make sense intuitively. I especially like the idea of distilling the outcome of the planning procedure to the goal-conditioned policy by self-imitation. This technique provides a way for the agent to bootstrap its own learning.
* The empirical evaluation is thorough and solid. PIG demonstrates strong performance against baseline methods in the experiments. The ablation studies further demonstrate the efficacy of both proposed techniques.
* I appreciate that the discussion on limitations of PIG at the end of the paper.

# Weakness
* The authors mainly discuss related works in the goal-conditioned reinforcement learning literature. However, I believe the idea of distilling the planner's outcome into the goal-conditioned policy is connected to some works in the broader planning context. For example, AlphaGoZero [2] distills the outcome of the Monte-Carlo tree search (MCTS) planning procedure into a prior policy. Similarly, SAVE [1] distills the MCTS outcomes into the action-value function. I strongly suggest the authors to broaden the scope of the related work section and add a discussion on the aforementioned works.
* It would be good if the ablated versions of PIG are also compared to the baseline methods. For example, by reading the numbers approximately from Figure 4 and Figure 7, it looks like PIG w/o skipping performs worse than MSS. This observation would be intriguing if true because I would expect the self-imitation learning technique itself to be beneficial.

# Questions and discussion
* In Equation (4), all intermediate subgoals are equally weighted. But the idea behind the subgoal skipping technique essentially says the opposite. I wonder if we can further improve PIG by using a similar idea as subgoal skipping and adaptively weighting the subgoals in Equation (4). I would love to hear the authors' comment.
* Why did you choose U-shaped Ant Maze for the experiment in Figure 5 instead of the larger version of it? PIG shows the largest improvement in large U-shaped AntMaze in Figure 4. In the current Figure 5, MSS + PIG w/o planner seems to perform equally to the MSS. I am very curious to see if it can actually perform better than MSS in a more challenging environment.
* Despite providing recommendations on how to choose $\beta$ automatically, the values are fixed (by manually tuning I suppose?) in the experiments. Why is it?

# Minor suggestions
* In the second paragraph of Section 3.3, perhaps replace the edge notation ${l^{1}, l^{2}}$ by $<l^{1}, l^{2}>$.
* In the fourth line in "Evaluation" paragraph of Section 5.1, I think you mean Figure 4, not Table 4.
* Figure 6 appears before Figure 5. Please consider fixing it.

**References**
1. Hamrick _et al_ 2020, Combining Q-Learning and Search with Amortized Value Estimates
2. Silver _et al_ 2017, Mastering the Game of Go without Human Knowledge

**Summary Of The Paper:**

This work addresses goal-conditioned reinforcement learning and proposes two techniques to improve existing graph-based planning methods. The first technique is to distill the outcome of the graph-based planning procedure into the goal-conditioned policy. The second technique is to randomly skip subgoals along the planned path when the agent interacts with the environment. The authors name the combination of these two techniques PIG and evaluate PIG in a set of continuous control tasks. Empirical results show that PIG outperforms strong baselines and improve performance in all tasks. The authors also provide practical recommendations on setting the hyperparameters introduced by PIG.

**Summary Of The Review:**

This paper proposes two techniques to improve graph-based planning methods in goal-conditioned reinforcement learning. The proposed techniques are thoroughly evaluated and demonstrate strong empirical performance against baselines. The paper is well-written and the method is clearly presented. Thus I recommend acceptance.

---

> ### Author Response · Authors · 2022-11-11
> **Response to reviewer 4xSo (1/2)**
>
> Dear reviewer 4xSo,
>
> We express our deep appreciation for your time and insightful comments. We address your comments one by one. The revisions made are marked with “$\text{\color{magenta}magenta}$” in the revised paper.
>
> ---
>
> **Q1: The authors mainly discuss related works in the goal-conditioned reinforcement learning literature. However, I believe the idea of distilling the planner's outcome into the goal-conditioned policy is connected to some works in the broader planning context. For example, AlphaGoZero [2] distills the outcome of the Monte-Carlo tree search (MCTS) planning procedure into a prior policy. Similarly, SAVE [1] distills the MCTS outcomes into the action-value function. I strongly suggest the authors to broaden the scope of the related work section and add a discussion on the aforementioned works.**
>
> **A1.** Thank you for the suggestion. AlphaGoZero [2] and SAVE [1] have an interesting connection to our framework in that they distill behavior from a planner into a policy. Specifically, as you mentioned, AlphaGoZero and SAVE distill MCTS outcomes into a prior policy and the action-value function, respectively, and PIG distills planned-subgoal-conditioned policies into the target-goal-conditioned policy. We expect that the learning scheme of the distilling planner into a policy would give insights into developing algorithms for a broader range of fields in the future. Following your suggestion, we broaden the scope of the related work section and add a discussion in the revised paper.
>
> ---
>
> **Q2: It would be good if the ablated versions of PIG are also compared to the baseline methods. For example, by reading the numbers approximately from Figure 4 and Figure 7, it looks like PIG w/o skipping performs worse than MSS. This observation would be intriguing if true because I would expect the self-imitation learning technique itself to be beneficial.**
>
> **A2.** To address your comment, we compare PIG w/o skipping and MSS and find that PIG w/o skipping performs better than MSS in Pusher and L-shaped AntMaze and is on par with MSS in 2DReach, as the table below shows. These results support that both of our components, i.e., self-imitation loss and subgoal skipping, are beneficial. We update Figure 7 by adding learning curves of MSS in the revised paper.
>
> ###### $\textbf{Mean (standard deviation) of success rate on 2DReach}$
> \begin{array}{lcccc}
> \text{Timesteps}  & 0.25 * 10^5 & 0.5 * 10^5 & 0.75 * 10^5 & 1.0 * 10^5 \newline
> \hline
> \textbf{MSS + PIG with skipping} &  \bf{0.96} \\; (0.03) & \bf{0.95} \\; (0.08) & \bf{0.97} \\; (0.03) & \bf{0.95} \\; (0.05) & \newline
> \text{MSS + PIG without skipping} & 0.63 \\; (0.15) & 0.77 \\; (0.09) & 0.83 \\; (0.06) & 0.86 \\; (0.03) & \newline
> \text{MSS} & 0.62 \\; (0.06) & 0.79 \\; (0.12) & 0.82 \\; (0.09) & 0.83 \\; (0.03) & \newline
> \end{array}
>
> ###### $\textbf{Mean (standard deviation) of success rate on Pusher}$
> \begin{array}{lccc}
> \text{Timesteps}  & 1.0 * 10^5 & 2.0 * 10^5 & 3.0 * 10^5 \newline
> \hline
> \textbf{MSS + PIG with skipping} & \bf{0.55} \\; (0.04) & \bf{0.83} \\; (0.10) & \bf{0.92} \\; (0.05) & \newline
> \text{MSS + PIG without skipping} & 0.17 \\; (0.13) & 0.47 \\; (0.17) & 0.57 \\; (0.05) & \newline
> \text{MSS} & 0.07 \\; (0.01) & 0.07 \\; (0.02) & 0.14 \\; (0.04) & \newline
> \end{array}
>
> ###### $\textbf{Mean (standard deviation) of success rate on L-shaped AntMaze}$
> \begin{array}{lccccc}
> \text{Timesteps}  & 1.0 * 10^5 & 2.0 * 10^5 & 3.0 * 10^5 & 4.0 * 10^5  & 5.0 * 10^5 \newline
> \hline
> \textbf{MSS + PIG with skipping} & \bf{0.18} \\; (0.11) & \bf{0.62} \\; (0.08) & \bf{0.80} \\; (0.10) & \bf{0.82} \\; (0.13) & \bf{0.84} \\; (0.12) & \newline
> \text{MSS + PIG without skipping} & 0.12 \\; (0.08) & 0.50 \\; (0.15) & 0.66 \\; (0.06) & 0.77 \\; (0.09) & 0.83 \\; (0.04) & \newline
> \text{MSS} & 0.10 \\; (0.03) & 0.39 \\; (0.09) & 0.59 \\; (0.11) & 0.53 \\; (0.12) & 0.72 \\; (0.05) & \newline
> \end{array}
>
> ---
>
> **Q3: In Equation (4), all intermediate subgoals are equally weighted. But the idea behind the subgoal skipping technique essentially says the opposite. I wonder if we can further improve PIG by using a similar idea as subgoal skipping and adaptively weighting the subgoals in Equation (4). I would love to hear the authors' comment.**
>
> **A3.** Thank you for the insightful comment. As you mentioned, one can further improve PIG by adaptively weighting the subgoals for Equation (4): self-imitation loss. For example, one could assign higher weights to nearby subgoals when the policy is inaccurate for faraway subgoals (during the early stages of training). This adaptive weighting would be an interesting future direction. Thank you!

---

> > ### Author Response · Authors · 2022-11-11
> > **Response to reviewer 4xSo (2/2)**
> >
> > **Q4: Why did you choose U-shaped Ant Maze for the experiment in Figure 5 instead of the larger version of it? PIG shows the largest improvement in large U-shaped AntMaze in Figure 4. In the current Figure 5, MSS + PIG w/o planner seems to perform equally to the MSS. I am very curious to see if it can actually perform better than MSS in a more challenging environment.**
> >
> > **A4.** To address your comment, we provide experimental results in the large version of the U-shaped AntMaze for the experiment in Figure 5. In large U-shaped AntMaze, we also find that training with PIG enables successfully reaching the target-goal even without the planner at test time. Intriguingly, after $5 \times 10^5$ environment steps, a policy trained by our approach performs better even without access to the planner at test time compared to MSS, which uses the planner at test time. The results and related discussions are added in Supplemental material C.5 in the revised paper.
> >
> > ###### $\textbf{Mean (standard deviation) of success rate on large U-shaped AntMaze}$
> > \begin{array}{lcccc}
> > \text{Timesteps}  & 2.5 * 10^5 & 5.0 * 10^5 & 7.5 * 10^5 & 10.0 * 10^5 & \newline
> > \hline
> > \textbf{MSS + PIG (Ours)} & \bf{0.09} \\; (0.07) & \bf{0.39} \\; (0.09) & \bf{0.32} \\; (0.10) & \bf{0.48} \\; (0.14) & \newline
> > \text{MSS} & 0.04 \\; (0.04) & 0.10 \\; (0.07) & 0.17 \\; (0.10) & 0.24 \\; (0.10) & \newline
> > \hline
> > \textbf{MSS + PIG w/o planner (Ours)} & 0.00 \\; (0.00) & \bf{0.03} \\; (0.03) & \bf{0.25} \\; (0.19) & \bf{0.32} \\; (0.19) & \newline
> > \text{MSS w/o planner} & 0.00 \\; (0.00) & 0.00 \\; (0.00) & 0.00 \\; (0.00) & 0.00 \\; (0.00) & \newline
> > \end{array}
> >
> > ---
> >
> > **Q5: Despite providing recommendations on how to choose $\lambda$ automatically, the values are fixed (by manually tuning I suppose?) in the experiments. Why is it?**
> >
> > **A5.** As you mentioned, we have recommended a way to automatically set the balancing coefficient $\lambda$ to guide researchers when they extend our work into new environments in the future. We note that the performance with automatic $\lambda$ was better than the baseline (MSS) in Figure 11 of the original draft, but we used a fixed balancing coefficient because fixed $\lambda$ was better than automatic $\lambda$ (and also than the MSS) as the table below shows. We hypothesize that this is because using fixed $\lambda$ can be more stable in training than our suggestion of automatic lambda. Nevertheless, it would be interesting future work to develop an automatic setting of $\lambda$, which is better than a fixed one.
> >
> > ###### $\textbf{Mean (standard deviation) of success rate on 2DReach}$
> > \begin{array}{lcccc}
> > \text{Timesteps}  & 0.25 * 10^5 & 0.5 * 10^5 & 0.75 * 10^5 & 1.0 * 10^5 & \newline
> > \hline
> > \textbf{MSS + PIG with }\lambda = 1 & \bf{0.96} \\; (0.03) & \bf{0.95} \\; (0.08) & \bf{0.97} \\; (0.03) & \bf{0.95} \\; (0.05) \newline
> > \textbf{MSS + PIG with automatic } \lambda & 0.77 \\; (0.14) & 0.86 \\; (0.13) & 0.92 \\; (0.15) & 0.94 \\; (0.11) \newline
> > \text{MSS} & 0.63 \\; (0.15) & 0.77 \\; (0.09) & 0.83 \\; (0.06) & 0.86 \\; (0.03) \newline
> > \end{array}
> >
> > ###### $\textbf{Mean (standard deviation) of success rate on L-shaped AntMaze}$
> > \begin{array}{lccccc}
> > \text{Timesteps}  & 1.0 * 10^5 & 2.0 * 10^5 & 3.0 * 10^5 & 4.0 * 10^5 & 5.0 * 10^5 \newline
> > \hline
> > \textbf{MSS + PIG with }\lambda = 1 & \bf{0.14} \\; (0.08) & \bf{0.68} \\; (0.10) & \bf{0.77} \\; (0.05) & \bf{0.78} \\; (0.07) & 0.72 \\; (0.15) \newline
> > \textbf{MSS + PIG with automatic } \lambda & 0.09 \\; (0.06) & 0.36 \\; (0.16) & 0.71 \\; (0.12) & 0.74 \\; (0.08) & \bf{0.83} \\; (0.10) \newline
> > \text{MSS} & 0.10 \\; (0.03) & 0.39 \\; (0.09) & 0.59 \\; (0.11) & 0.53 \\; (0.12) & 0.72 \\; (0.05) \newline
> > \end{array}
> >
> > ---
> >
> > **Q6: Minor suggestion: editorial comment**
> >
> > **A6.** Thank you for the comment. Following your comment, we replace the edge notation and fix the typo in the revised paper. We plan to change Figure IDs in the final draft. Thank you!

---

### Official Review · Reviewer_tgwg · 2022-10-24

**Confidence:** 3
**Correctness:** 4
**Technical Novelty And Significance:** 3
**Empirical Novelty And Significance:** 3
**Recommendation:** 6

**Clarity, Quality, Novelty And Reproducibility:**

The paper is clearly written and well motivated. To the best of my knowledge the proposed new objective is novel and leads to empirical performance improvements. I have no concerns about reproducibility since the method is relatively straightforward and there is code provided.

**Strength And Weaknesses:**

Strengths:
* well motivated new objective that empirically seems to improve performance
* clearly written paper with pretty thorough evaluation

Weaknesses:
* since PIG can be applied on top of L3P it might be nice to see those results as well.
* in some of the learning curves not all (or even no) methods seem to have converged. It would be interesting to evaluate for longer to be able to assess whether the empirical improvements are in learning speed or also in asymptotic performance.
* the specific implementation of subgoal skipping seems a little ad-hoc. Is it obvious that this is the right choice? I think mentioning the comparison to random goal sampling along the path in the main text might be good. I would also like to see an ablation over a wider range of $\alpha$ (fig 11).
* It's not clear to me that any of these approaches, even with the proposed improvements would scale to higher dimensional tasks.
* figure 8: which environment is this?
* evaluation: are you reporting standard deviation or standard error in the mean?

**Summary Of The Paper:**

This paper focusses on the setting of goal-conditioned RL where subgoals are provided by graph based planners. It introduces two algorithmic contributions:
* a self imitation loss that is inspired by the "optimal substructure property" ie the fact that parts of an optimal path are themselves optimal. This idea is used to regularize the policy conditioned on a final goal towards those conditioned on the subgoal along a planner path
* subgoal skipping: instead of always attempting to reach the first subgoal the policies sometimes directly attempts more distant subgoals along a planned path to aid with exploration.
The proposed innovations are primarily evaluated in a number of maze environments. Empirically the proposed approach compares favourably to baselines, especially in more complex maze environments.

**Summary Of The Review:**

Nice paper with a well-motivated simple new objective that appears to lead to improved performance in goal-conditioned RL with graph-based planner guidance.

---

> ### Author Response · Authors · 2022-11-11
> **Response to reviewer tgwg**
>
> Dear review tgwg,
>
> We express our deep appreciation for your time and insightful comments. We address your comments one by one. The revisions made are marked with “$\text{\color{magenta}magenta}$” in the revised paper.
>
> ---
>
> **Q1: PIG on top of L3P would be nice to see as well.**
>
> **A1.** We have already applied PIG on top of L3P, as shown in Figure 9 of Supplemental material C.1 and mentioned in the main text (Section 5.2) in our original draft. This supports our claim that PIG is a general method that can improve the sample-efficiency of graph-based GCRL methods.
>
> ---
>
> **Q2: longer evaluation to assess whether the empirical improvements are in learning speed or also in asymptotic performance.**
>
> **A2.** To address your comment, we evaluate PIG and MSS with extended timesteps (i.e., from $10 \times 10^5$ to $30 \times 10^5$ on Large U-shaped AntMaze). As shown in the Table below, we find that PIG can improve both sample-efficiency and asymptotic performances of MSS. This shows that enhanced policy learning via information distillation from the planner can also improve asymptotic performance. We add these results in Supplementary material C.8 of the revised draft.
>
> ###### $\textbf{Mean (standard deviation) of success rate on Large U-shaped AntMaze.}$
> \begin{array}{lccc}
> \text{Timesteps}  & 10.0 * 10^5 & 20.0 * 10^5 & 30.0 * 10^5 \newline
> \hline
> \textbf{MSS + PIG (Ours)} & \bf{0.61} \\; (0.12) &  \bf{0.61} \\; (0.14) & \bf{0.75} \\; (0.14) & \newline
> \text{MSS} & 0.31 \\; (0.08)  & 0.28 \\; (0.22) &  0.48 \\; (0.18) & \newline
> \end{array}
>
> ---
>
> **Q3: the specific implementation of subgoal skipping seems a little ad-hoc. Is it obvious that this is the right choice? I think mentioning the comparison to random goal sampling along the path in the main text might be good. I would also like to see an ablation over a wider range of alpha (fig 11).**
>
> **A3.** Our subgoal skipping strategy is designed to provide an appropriate subgoal because (1) a subgoal is sampled from the searched shortest path, and (2) it is not likely to choose a farther subgoal if a policy is not trustworthy for faraway subgoals. Specifically, for (1), all the sampled subgoals are on the way to the final goal, so they are proper waypoints to achieve the final goal. For (2), thanks to our skipping design, we rarely skip subgoals if a policy is not trustworthy for faraway subgoals. We mention the comparison to random goal sampling along the path in 4.2 of the revised paper with related discussion.
>
> To address your comment, we conduct experiments with a wider range of alpha (i.e., [0.1, 100] rather than [1, 20]) and update Figure 11-c. We observe that even in the wider range of alpha, using subgoal skipping makes a significant gain compared to the baseline, which does not use subgoal skipping (i.e., $\alpha = 0$). We note that performance is robust in a wide range of $\alpha$ (i.e., [1, 100]).
>
> ###### $\textbf{Mean (standard deviation) of success rate on Pusher}$
> \begin{array}{lccc}
> \text{Timesteps}  & 1.0 * 10^5 & 2.0 * 10^5 & 3.0 * 10^5 \newline
> \hline
> \alpha = 100 & \bf{0.74} \\; (0.14) &  0.78 \\; (0.11) & 0.89 \\; (0.07) & \newline
> \alpha = 10 & 0.65 \\; (0.15) & \bf{0.88} \\; (0.08) & \bf{0.93} \\; (0.09) & \newline
> \alpha = 1 & 0.66 \\; (0.04) & 0.83 \\; (0.10) & 0.92 \\; (0.05) & \newline
> \alpha = 0.1 & 0.26 \\; (0.08) & 0.59 \\; (0.14) & 0.68 \\; (0.21) & \newline
> \alpha = 0 & 0.09 \\; (0.04) & 0.06 \\; (0.03) & 0.12 \\; (0.05) & \newline
> \end{array}
>
> ---
>
> **Q4: It's not clear to me that any of these approaches, even with the proposed improvements, would scale to higher dimensional tasks.**
>
> **A4.** Graph-based planning for GCRL has been scaled to high-dimensional tasks (i.e., visual navigation) [1,2,3] (with an additional encoder such as ResNet-18), and PIG is also applicable to high-dimensional state spaces because our algorithmic components (self-imitation loss and subgoal skipping) do not depend on the dimension of state spaces. It would be interesting future work to extend our work into more high-dimensional observation space. We add a related discussion in Section 6 of the revised paper.
>
> **References**
>
> [1] Nikolay Savinov, Alexey Dosovitskiy, Vladlen Koltun, “Semi-parametric topological memory for navigation”, ICLR 2018.
>
> [2] Benjamin Eysenbach, Ruslan Salakhutdinov, Sergey Levine, “Search on the Replay Buffer: Bridging Planning and Reinforcement Learning”, NeurIPS 2019.
>
> [3] Christopher Hoang, Sungryull Sohn, Jongwook Choi, Wilka Carvalho, Honglak Lee, “Successor Feature Landmarks for Long-Horizon Goal-Conditioned Reinforcement Learning”, NeurIPS 2021.
>
> ---
>
> **Q5: figure 8: which environment is this?**
>
> **A5.** For Figure 8, we used L-shaped AntMaze. We clarify this in the revised paper.
>
> ---
>
> **Q6: evaluation: are you reporting standard deviation or standard error in the mean?**
>
> **A6.** We used standard deviation for all the experiments as mentioned in the Evaluation paragraph of Section 5.1 in the original draft.

---

### Author Response · Authors · 2022-11-11
**General response**

Dear reviewers and AC,

We deeply appreciate your time and effort in reviewing our draft.

Our work firstly brings optimal substructure property into GCRL literature and proposes self-imitation loss and subgoal skipping as our technical contribution. We are delighted to find that reviewers highlighted our well-motivated and intuitive idea (reviewer tgwg, 4xSo), strong empirical performance with thorough evaluation (all), and clear write-up (all)

In response to the questions and concerns reviewers raised, we have carefully revised and improved the draft with the following additional experiments and discussions:

- Explaining related work in more detail and discussing work that distills planning into a policy (Section 2).
- Adding further details about graph construction (Section 3.3 and Supplemental material D.1).
- Clarification on why our subgoal skipping provides appropriate subgoals (section 4.2).
- Clarification on how our approaches can be scaled up to high-dimensional state spaces (section 6).
- Additional experiments with a larger maze without a planner in evaluation (Supplemental material C.5)
- Additional experiments with stochastic transition model (Supplemental material C.6).
- Additional ablation studies with more environments (Supplemental material C.7).
- Additional experiments with extended timesteps (Supplemental material C.8).
- Clarification about hyperparameter search for ours and baselines (Supplemental material D.2)

These updates are temporarily highlighted in “$\text{\color{magenta}magenta}$” for your convenience to check.

We also appreciate your continued effort to provide further feedback until the very end of the response/discussion phase. We will make sure to reflect on the comments in the final version.

Thank you very much,

Authors.

Thank you

---

### Decision · Program_Chairs · 2023-01-20

**Decision:**

Accept: poster

**Justification For Why Not Higher Score:**

Overall, I think this paper foregrounds some interesting ideas.  I personally share the concern raised by reviewer tgwg that this class of approach might be a bit limited in the problems it is applicable to, as well as the adjacent concern by reviewer X3ba questioning how general the approach is (i.e., restriction to goal-reaching tasks).  However, within the problem-space that this paper seeks to solve, there is a clear contribution.

**Justification For Why Not Lower Score:**

4/4 reviewers see a clear and meaningful enough contribution, and I agree.

**Metareview: Summary, Strengths And Weaknesses:**

The contributions of this paper involve leveraging the optimal substructure property to better train goal-conditioned policies and performing subgoal skipping during training and deployment to allow for discovering more efficient plans.

The reviewers generally found the paper well motivated and clear, with initially mixed opinions about the thoroughness of the paper.  Reviewer tgwg expressed some concern that the approach may not scale well to higher-dimensional problems.  The authors replied that graph-based planning has been applied to e.g. visual perception problems, so they are optimistic.  Reviewer 4xSo found the evaluation sufficiently thorough, but pointed to some connections in the existing literature that the authors could include.  Reviewer X3ba initially gave the paper a 3 due to potentially limited applicability.  However, they updated their score to a 6 based on author explanation and additional experiments.  Reviewer syAR's biggest concern was related to baseline comparisons, especially around hyperparameter selection.

Generally, the authors made an effort to address reviewer concerns to a reasonable standard.

I'm comfortable endorsing the reviewer consensus for this paper to be accepted.

**Note From Pc:**

if the above contains the word "oral" or "spotlight" please see: "oral" presentation means -> notable-top-5% and "spotlight" means -> notable-top-25%. As stated in our emails, we are disassociating presentation type from AC recommendations